# Modeling soil organic carbon dynamics in temperate forests ~~using~~ with Yasso07

Zhun Mao[1,8*], Delphine Derrien[1], Markus Didion[2], Jari Liski[3,9], Thomas Eglin[4], Manuel Nicolas[5], Mathieu Jonard[6], Laurent Saint-André[1,7]

[1] INRA, UR BEF – Biogéochimie des Ecosystèmes Forestiers, 54280 Champenoux, France

[2] Swiss Federal Institute for Forest, Snow and Landscape Research WSL, 8903 Birmensdorf, Switzerland

[3] Finnish Environment Institute, Ecosystem Change Unit, Natural Environment Centre, Mechelininkatu 34a, P.O.Box 140, 00251 Helsinki, Finland

[4] ADEME – DPED – Service Agriculture et Forêts, 49004 Angers, France

[5] Office National des Forêts, Direction Forêts et Risques Naturels, Département Recherche~~ et~~ Développement-Innovation~~ Bâtiment B~~, Boulevard de Constance, 77300 Fontainebleau, France

[6] Université Catholique de Louvain, Earth and Life Institute, Croix du Sud 2, L7.05.09, 1348 Louvain-la-Neuve, Belgium

[7] CIRAD, UMR ECO&Sols, place Viala, 34398 Montpellier Cedex 5, France

[8] Amap, Inra, University Montpellier, Cnrs, Ird, Cirad, Montpellier, France

[9] Finnish Meteorological Institute, P.O. Box 503, 00101 Helsinki, Finland

*Corresponding author: Zhun Mao; email address: maozhun04@126.com.

**Abstract.** ~~Facing~~ In a context of global changes, modeling and predicting the dynamics of soil carbon stocks in forest ecosystems is vital but challenging. Yasso07 is considered ~~as~~ one of the most promising models for such a purpose. We ~~aim at examining~~examine the accuracy of its prediction of ~~the~~ soil carbon dynamics over the whole French metropolitan territory at a decennial time scale.

We used data from 101 sites ~~of~~in the RENECOFOR network, which encompasses most of the French temperate forests. These data include (i) ~~yearly measured~~ the quantity of aboveground litterfall from 1994 to 2008, measured yearly; and (ii) the soil carbon stocks measured twice at an interval of approximately~~c.a.~~ 15 years (once in the early 1990s ~~versus~~and around 2010). ~~Using Yasso07, we simulated~~We used Yasso07 to simulate the annual changes in carbon stocks ~~changes~~ (tC ha$^{-1}$ yr$^{-1}$) ~~per~~for each site and then compared ~~them~~the estimates with ~~the~~ ~~actual measured~~recorded ~~ones~~data. We carried out meta-analyses to reveal the variability in litter biochemistry ~~between~~ in different tree organs for conifers and broadleaves. We also performed sensitivity analyses to explore Yasso07's sensitivity to annual litter input~~s~~ and model initialization setting~~s~~.

At the national level, the simulated changes in the annual carbon stock ~~changes~~ (ACC, +0.00 ± 0.07 tC ha$^{-1}$ year$^{-1}$, mean ± standard error) ~~stayed in~~were of the same order of magnitude as the observed ones (+0.34 ± 0.06 tC ha$^{-1}$ year$^{-1}$). However, t~~T~~he correlation between predicted and measured ACC remained weak (R² <0.1). There was significant overestimation for broadleaved stands and underestimation for conifer~~ous~~s sites. Sensitivity analyses showed that the final estimated carbon stock was strongly affected by settings in the model initialization,

including litter and soil carbon quantity and quality, and also by simulation length. Carbon quality set with the partial steady-state assumption gave a better ~~model~~ fit than ~~that~~ the model with the complete steady-state assumption.

~~Taking~~ With Yasso07 as the support model ~~support~~, we ~~revealed the current~~ showed that there is currently a bottleneck ~~of~~ in soil carbon modelling and prediction due to ~~lacking~~ a lack of knowledge or data on soil carbon quality and fine root quantity in the litter.

**Nomenclature and abbreviations**

| Name | Meaning |
|---|---|
| carbon stock (CS) | Quantity of ~~soil~~ organic carbon stocked in the soil (in tC ha$^{-1}$) |
| carbon stock change | Increment (positive value) or decrement (negative value) of ~~soil~~ organic carbon stocked in the soil ~~from~~ between ~~the~~ year t1 ~~to~~ and ~~the~~ year t2 (in tC ha$^{-1}$) |
| annual carbon stock change (ACC) | Changes in c~~c~~arbon stocks per year (in tC ha$^{-1}$ year$^{-1}$) |
| carbon pools | The Yasso07 model contains a series of organic compounds involved in decomposition processes differing in solubility ~~in solvents~~ and mean residence time ~~in decomposition processes~~: water soluble compounds (W), acid-hydrolysable compounds (A); non-polar solvent, ethanol or dichloromethane compounds (E), and non-soluble and non-hydrolyzable compounds (N). For soil, there is an ~~extra~~ additional recalcitrant pool ~~named~~ called "humus" (H). Note: in this paper, "N" only denotes non-soluble and non-hydrolyzable compounds; nitrogen is spelled in full ~~letter~~ when mentioned. |
| coarse woody litter | Litter ~~yield~~ originating from either coarse aboveground residues due to either harvests or storms (including coarse branches~~, de fined as branched~~ of >4 cm in diameter and miscellaneous residues) and coarse roots ~~(defined as those~~ of >5 mm in diameter~~)~~ |
| fine non-woody litter | Litter ~~yield~~ originating from either natural above-ground litterfall (leaves, small branches) or fine root~~s~~ activities |
| litter carbon quality | ~~Composition of litter~~Litter carbon (in %) belonging to the A, W, E and N carbon pools ~~(in %)~~(see "carbon pools" above) |
| litter quantity | Annual litter ~~input~~ accumulation (in tC ha$^{-1}$ year$^{-1}$) |
| soil carbon quality | ~~Composition of soil~~Soil carbon (in %) belonging to the A, W, E, N and H carbon pools ~~(in %)~~ |

**1 Introduction**

The ~~current global~~ carbon stock in ~~global~~ soils, including forest litter and peatlands, is 1500 to 2400 GtC, and thus greatly exceed~~s stock~~s ~~ing that~~ in vegetation, found mainly in forests (350 to~~à~~ 550 GtC~~, mainly in forests~~), and in the atmosphere (829 GtC in 2011, IPCC, 2014). Soils share a common interface with all the other spheres and play a key role in driving the global carbon cycle. Soil carbon stock dynamics are directly related to the greenhouse gas emissions (notably carbon dioxide; $CO_2$) that are leading to the global warming effect (IPCC, 2014). An accurate estimation of soil carbon stock dynamics ~~allows~~ would allow us to better understand the turnover rate and fate of soil carbon flux at both local and global geographical scales. ~~Facing~~ In the context of global changes, ~~this task~~ accurate estimation is essential ~~for the evaluation of the~~ to evaluate the climate change mitigation potential~~s~~ of forests and ~~the~~ to support ~~of~~ environmental policy decisions.

Significant challenges exist ~~for accurate estimation of~~ when attempting to accurately estimate changes in soil carbon stock~~s changes~~. Current soil monitoring networks are generally not able to detect changes on timescales of less than 10 years (Saby et al. 2008). To obtain estimates for changes in soil C stock~~s change estimates~~ at shorter intervals, ~~such~~ as is, for example, required for ~~the~~ annual reporting to the United Nations Framework Convention on Climate Change and the Kyoto Protocol, ~~the use of~~ using models is encouraged (IPCC, 2011). Numerous models have been elaborated ~~for evaluating~~ to evaluate soil carbon dynamics (Manzoni and Porporato, 2009). The vast majority of terrestrial soil carbon models developed at the global or ~~at~~ the plot scale~~s~~, (e.g., CENTURY (Parton et al**.,** 1987), RothC (Coleman and Jenkinson, 1996) and ORCHIDEE (Krinner et al., 2005))~~,~~ assume that decomposition is the first order decay~~ing~~ process, which accounts ~~accounting~~ for the size of soil carbon pools~~.,~~ ~~despite the existence of~~ However, ~~criticism to this~~ assumption has been criticized and it has been argued, ~~arguing~~ that a priming effect and the associated ~~induced~~ carbon pool interactions should also be considered in model algorithms (Wutzler and Reichstein, 2013). The dynamics of carbon pools depend on the quantity and quality of litter inputs and on temperature, soil moisture and other soil parameters, e.g. texture, structure, chemical richness, pH etc. (Todd-Brown et al., 2012). Incorporating explicit mechanisms such as microbial activities or carbon protection by the soil matrix into soil carbon models has repeatedly been suggested in ~~the last~~ recent years (Schmidt et al., 2011; Lehmann and Kleber, 2015). However, for forest ecosystems, ~~such~~ refined mechanistic input data often remain ~~often~~ limited. Accordingly, the typical time-step for litter input demanded by most ~~of~~ forest soil carbon models ~~for forests~~ is year, ~~not~~ rather than month (but see RothC, Coleman and Jenkinson, 1996) or day (but see

Romul, Chertov et al., 2001) (Didion et al., 2016). At this yearly- timescale, it is common to consider microbial communities and processes as ~~a~~ relatively stable factor~~s~~ (Todd-~~b~~Brown et al, 2012)~~;,~~ ~~and~~ in this case, the assumption ~~of~~ that carbon dynamics are governed by first order decay may therefore be reasonable.

This is the choice made by the group who built the Yasso ~~model~~ (Liski et al., 2005) and Yasso07 (Tuomi et al., 2009; 2011a and 2011b) model~~s~~. ~~(Tuomi et al., 2009; 2011a and 2011b), i.e.~~Yasso07 is an improved version of Yasso with more refined carbon pooling and abundant data for calibration. The ~~intention of the~~ models' developers' intention was~~is~~ to ~~let~~ make their models ~~be~~ suitable for general forestry applications by taking into account the ~~low~~ limited availability of forest soil and litter data (Liski et al., 2005). Yasso07 explicitly defines several ~~chemical~~ pools of chemical compounds in litter carbon (Tuomi et al., 2011b) and possesses well-defined, biologically meaningful and measurable parameters. ~~Due~~Thanks to these qualities, Yasso ~~and~~ or Yasso07 ~~were~~ has been applied in more than 70 case studies (URL: http://www.syke.fi/en-US/Research__Development/Research_and_development_projects/Projects/Soil_carbon_model_Yasso/) in forest ecosystems in the northern hemisphere with generally high satisfaction levels ~~in comparison~~when compared with measured carbon values (e.g. Karhu et al., 2011 ; Rantakari et al., 2012; Ortiz et al., 2013 ; Didion et al., 2014; Lu et al., 2015; Wu et al., 2015). Yet, so far most of these applications have been limited to local case studies, especially ~~those on~~in cold forests with limited tree species diversity (e.g. boreal or montane forests). Rarely have previous studies validated Yasso07 based on data (i) ~~of~~ from long-term observations (here defined as ~~data of~~ >10 years), (ii) from temperate forests with a much higher diversity of tree species, or (iii) on changes in carbon stock~~s~~ ~~changes~~ (in tC ha$^{-1}$ year$^{-1}$). This is partially due to the lack of extensive long--term soil carbon monitoring in forest ecosystems, which differ in climatic and soil conditions and species and~~,~~ stretch over ~~a~~ large territorial scale~~s~~. Nevertheless, Yasso07 ~~has been~~is considered ~~as~~ to be one of the potentially appropriate models ~~appropriate~~ for evaluating national and continental inventories of the forest carbon balance in Europe (Hernández et al. 2017). It is therefore of ~~high~~ considerable interest to assess the ability of Yasso07 to reflect the carbon balance in different European forest ecosystems at large spatial-temporal scales. Moreover, as a carbon pool--based model, Yasso07 shares certain ~~similar~~ principles ~~to~~ with other prevailing soil carbon models in the same genre (e.g., RothC, CENTURY etc.). Applying Yasso07 as an example model in this case study, ~~we~~ may also allow us to ~~learn from this application case~~improve ~~for~~ future carbon modelling for temperate forests in general.

The ~~of~~ recorded field data ~~of~~ for carbon stock~~s~~ and litter quantity dynamics from the RENECOFOR network (URL: http://www.onf.fr/renecofor/@@index.html), National Forest ~~Management Agency~~Office (ONF), France, offered us a valuable opportunity for model validation. The 101 forest sites included in this study~~considered from this network~~ are located all over the French metropolitan territory and cover the most common forest types and tree species. For each site, annual measurements of litterfall were available in addition to two inventories of soil organic carbon stock~~s~~ with an average interval of 15 years (minimum 12 years and maximum 20 years). These data allowed us to use site-specific observed soil carbon stock~~s~~ and above-ground litterfall dynamics as model input data.~~, thus reducing the uncertainties of~~ Approximations in ~~the~~ model input~~,~~ data ~~which were~~ have been identified as a major source of uncertaint~~yies~~ for ~~model~~ estimates in models for changes in ~~of~~ soil carbon stock~~s~~ ~~changes~~ (Ortiz et al. 2013). By ~~minimizing this source of uncertainty~~ensuring solid input data, we were able to minimize this source of uncertainty and~~to~~ focus on the inherent model structure.

~~Consistent with our objective~~We hope to contribute to the further development of soil carbon modeling~~, we aim at~~ by (i) testing and characterizing the ability of Yasso07 to model soil carbon stock dynamics for temperate forests, (ii) identifying limitations and providing suggestions for a better adaptation of the model for C dynamics in both deciduous and evergreen temperate forests, and (iii) discussing the perspectives based on the current state-of-the-art ~~of~~ in soil carbon modelling. Associated with the above aims, our null hypotheses are as follows: (i) Yasso07 predicts accurate and unbiased carbon stock changes at the national scal~~e;e~~ and (ii) the model's fit residuals (predicted data minus observed data) have null relationships with site characteristics (e.g. location, climate, forest type, soil type and initial carbon stock).

**2 Materials and methods**
**2.1 The ~~model~~ Yasso07 model**
The Yasso07 dynamic soil carbon model ~~Yasso07~~ is based on the general assumption that the
soil carbon stock is driven by the decomposition of different litter types, which may differ in
quantity and quality, and by climatic conditions. Litter carbon quality is represented by four
chemical compound groups ~~which have~~ with different decomposition rates (Tuomi et al.,
2009). Soil organic carbon is divided into these four relatively labile carbon pools and one
recalcitrant pool ~~named~~ called "humus" (H) (Fig. S1). The five pools differ in specific mass
loss rates and mass flows ~~among them~~. As in many other pool-based models, the H pool is
considered the oldest and most stable carbon pool, although recent studies ~~doubted~~ have
thrown doubt on its stability and even its physical existence ~~and stability~~ (see Lehmann and
Kleber, 2015). Some mass flows correspond to $CO_2$ release (microbial respiration). The mean
residence time of carbon in these pools varies, lasting ~~from~~ several months (i.e., water soluble
compounds, W), a few years (i.e., acid-hydrolysable compounds, A; non-polar solvent,
ethanol or dichloromethane compounds, E), several decades (i.e., non-soluble and non-
hydrolyzable compounds, N), or even several centuries (i.e., H).
Mathematically, the kernel equation of Yasso07 can be written as follows:
$$\dot{X}(t) = A_p K(c) X(t) + I(t)$$  (Eq. 1a)
where, symbols in bold capital letters ~~in bold~~ denote either vectors or matrices whilst those in
small letters in parentheses denote scalars; $X(t)$ is the vector describing the masses of the five
carbon pools (A, W, E, N, H) at time $t$; ~~and~~ $X(t)$ ~~are vectors~~ is the vector describing ~~describing~~
~~the masses of the five carbon pools (A, W, E, N, H) and~~ carbon mass changes in soil at time
~~($t$), respectively~~; $A_p$ is the mass flow matrix describing carbon allocation among pools; $K(c)$
is the decomposition matrix describing the decomposition rates as a function of climatic
conditions ($c$); and $I(t)$ is litter input to the soil, ~~with the last element~~ and is equal to 0 for the
last pool since, ~~as~~ "H" does not exist in litter~~s~~ form. (Eq. 1a) can be expressed in a more
detailed form:

$$
\begin{pmatrix} \partial x_A/\partial t \\ \partial x_W/\partial t \\ \partial x_E/\partial t \\ \partial x_N/\partial t \\ \partial x_H/\partial t \end{pmatrix} = \begin{pmatrix} -1 & p_{W \to A} & p_{E \to A} & p_{N \to A} & 0 \\ p_{A \to W} & -1 & p_{E \to W} & p_{N \to W} & 0 \\ p_{A \to E} & p_{W \to E} & -1 & p_{N \to E} & 0 \\ p_{A \to N} & p_{W \to N} & p_{E \to N} & -1 & 0 \\ p_{A \to H} & p_{W \to H} & p_{E \to H} & p_{N \to H} & -1 \end{pmatrix} \begin{pmatrix} k_A & 0 & 0 & 0 & 0 \\ 0 & k_W & 0 & 0 & 0 \\ 0 & 0 & k_E & 0 & 0 \\ 0 & 0 & 0 & k_N & 0 \\ 0 & 0 & 0 & 0 & k_H \end{pmatrix} \begin{pmatrix} x_A \\ x_W \\ x_E \\ x_N \\ x_H \end{pmatrix} + \begin{pmatrix} I_A \\ I_W \\ I_E \\ I_N \\ 0 \end{pmatrix}
$$

29                                                                        (Eq. 1b)

where $p_{F \rightarrow T}$ is the relative mass flow parameter between two pools (from $F$ to $T$; $F$ and $T$ can be any two pools among A, W, E, N and H) in the soil (dimensionless, $p_{F \rightarrow T} \in [0, 1]$). Temperature and precipitation are assumed not to affect mass flows $p$, but do influence mass loss rates $k_i$ ($i$ = A, W, E, N or H) according to:

$$k_i\ c = \alpha_i \exp \beta_1 T + \beta_2 T^2 [1 - \exp(\gamma P_a)] \qquad \text{(Eq. 2)}$$

where $\alpha_i$ is the mass loss rate parameter of the chemical pool $i$; and $\beta_1$, $\beta_2$ and $\gamma$ are parameters related to temperature ($T$, in °C) and precipitation ($P_a$, in mm).

To take into account the effect of litter size on litter decomposition rate, $k_i$ was multiplied by a litter size factor ($h_s$), which makes it possible to distinguish between different types of litters (e.g. foliage, coarse woody debris, stems, etc.) differing in diameter ($d$, in mm):

$$h_s\ d = \min (1 + \varphi_1 d + \varphi_2 d^2)^r, 1 \qquad \text{(Eq. 3)}$$

where $\varphi_1$, $\varphi_1$ and $r$ are parameters related to litter size.

Yasso07 has 44 parameters calibrated according to the Markov chain Monte Carlo (MCMC) method with the Metropolis-Hastings algorithm (Tuomi et al., 2011a). Currently, several calibrated parameter sets for Yasso07 are available, including the two most recent sets published by Tuomi et al. (2011) and Rantakari et al. (2012). Compared with the Rantakari 2012 set, the Tuomi 2011 set was calibrated using a wider range of observed foliage and root decomposition data. It is based on a combination of three sources of data: (i) a global dataset ($n > 9000$) of litterbags for mass loss of non-woody litters from approximately 100 sites in Europe and Northern and Central America covering a wide range of climate and soil conditions, forest types and tree species; (ii) a dataset ($n > 2000$) for mass loss of decomposing woody litter measured in Northern Europe; (iii) measured accumulation rates of soil carbon pools in forest sites along a 5300-year soil chronosequence in southern Finland to determine the residence time of the H carbon pool. The Tuomi 2011 parameter set contains 10,000 parameter vectors (each vector contains the values of all 44 Yasso07 parameters), which are randomly generated to take into account stochastic effect. In this study, we adapted the Tuomi 2011 set to the RENECOFOR dataset.

## 2.2 RENECOFOR network

The RENECOFOR network is part of the Level II network of the International Cooperative Program on Assessment and Monitoring of Air Pollution Effects on Forests (ICP Forests). The 101 sites (Fig. 1) considered in this study cover the most common types of forest

ecosystems in France, including even-aged forests in plains ~~area~~, pine plantations and uneven-aged mountain forests. They also ~~cover~~ host ~~the majority~~most of the tree species in France and central Europe, including *Quercus robur*. *Quercus petraea*, *Pseudotsuga Menziesii*, *Picea abies*, *Fagus sylvatica*, *Pinus pinaster*, *Pinus sylvestris* and *Abies alba*. At each forest site, annual ~~forest~~ woody and non-woody litter quantities ~~have been~~are either directly measured or estimated based on ~~the~~ existing dendrometric data.

**2.2.1 Soil carbon and soil physical and chemical properties**

At each site, soil carbon stocks (CS) were measured twice ~~with~~ at an interval of approximately 15 years (1993 – 95 for the first assessment and 2007 – 12 for the second one). At each site and for each assessment, soils ~~to a depth of 0.4 m~~ were sampled to a depth of 0.4 m ~~from~~ at five points selected in each of ~~the~~ five subplots and the samples were divided into different layers (0 – 0.1 m, 0.1 – 0.2 m and 0.2 – 0.4 m), including both organic and mineral soil layers. The temporal ~~evolution of the~~changes in soil CS ~~until~~ to a depth of 0.4 m was analyzed by Jonard et al. (2017). Composite samples were produced for each layer and subplot, ~~and~~ then analyzed for mass, bulk density, soil organic carbon and physical and chemical properties, including texture (~~percentages~~proportion of clay, silt and sand, in %), pH value, total nitrogen stock (in t ha$^{-1}$), carbon:nitrogen ratio (dimensionless), total phosphorus stock (in t ha$^{-1}$), and stocks of exchangeable aluminum (Al), calcium (Ca), potassium (K) and magnesium (Mg, in kmol ha$^{-1}$). We used the ~~s~~Soil physical and chemical propert~~yies~~ data measured during the first assessment (1993 – 95) ~~were used~~ for residual analyses (see Sect. 2.7)~~ and only those measured in the 1st inventories were used for this purpose.~~

Regarding the CS ~~of depth~~from 0.4 – 1.0 m in depth, only ~~the~~ data ~~of~~from the first assessment (1993 – 95) ~~are~~were available. Soil samples were obtained from only one soil profile per site at two mineral layers (0.4 – 0.8 m and 0.8 – 1.0 m). Bulk density and carbon concentrations measured at these layers were used to estimate soil carbon stock~~s~~ ~~until~~to a depth of 1.0 m. Table 2 ~~provides a synthesis of~~summarizes the data source for each of the 101 sites ~~of~~in the RENECOFOR network

~~-~~(URL: http://www.onf.fr/renecofor/sommaire/renecofor/reseau/20090119-130815-828957/@@index.html). More detailed information about each site and the soil sampling procedure is available in Supplementary Material I (Table S1) and Jonard et al. (2017).

### 2.2.2 Climate data

~~Necessary~~ The climate data required by Yasso07 includes annual mean precipitation (mm) and annual maximum, mean and minimum temperature~~s~~ (°C). These measured data were obtained from the ~~nearest~~ national Météo-France meteorological stations ~~of Météo-France~~ (http://www.meteofrance.com) ~~for~~ nearest each RENECOFOR site.

### 2.3 Litter quantity

Litter input (in tC ha$^{-1}$ yr$^{-1}$) comes from several sources (see Table 2)~~as follows~~. ~~The~~ We assumed a 0.5 conversion factor between biomass (dry matter) and carbon ~~was assumed to be 0.5~~ (Thomas and Martin, 2012).

Aboveground litter input from living trees includes leaves for broadleaves and needles for conifers, small branches, fruits and miscellaneous items (e.g., flower~~s~~, bud~~s,~~ etc.). Aboveground litterfall mass was ~~annually~~ measured annually between 1994 and 2008. For sites where litter quantity data from 1992 – 1993 and 2009 – 2012 were lacking, we used the mean litter quantity of all the other years ~~of~~ at the same site. The observed branch size in this aboveground category ~~is below~~ was less than 2 cm (fine branches). Branches and stems bigger than 2 cm due to natural mortality ~~should be~~were rare (~~as some of them~~since they can be salvaged) and ~~thus~~were therefore not included in our calculations. ~~Woody residues due to harvest or storms were estimated on the basis of repeated stand inventory data and species-specific height-girth and biomass.~~ Coarse woody litter input~~s~~ from harvesting residue~~s~~ or storms were estimated from full inventories performed by the ONF since 1991. Missing years of litter input ~~of~~ for this category ~~are~~ were gap-filled ~~using~~ with the average over the period. On average~~,~~ three~~3~~ years are missing per site ~~but~~ though there are ~~high~~ considerable differences amongst sites. The mode ~~is~~ was one year, and six~~6~~ sites ha~~d~~ve 10-11 missing years. ~~These residuals are assumed to be~~ We assumed the residues due to harvesting or storms would be coarse branches (> 4 cm in diameter, confirmed with the ONF) ~~as a function of~~based on aboveground tree characteristics. The quantities were estimated on the basis of repeated stand inventory data and species-specific height-girth relationships and biomass. Litter input from stems was set to 0, since in most cases~~,~~ stemwood was removed from the site after storm damage. Litter input from coarse woody roots ~~is~~ was considered to be equal to total root biomass, which ~~could be~~was estimated ~~using~~ through meta-analysis-based allometric equations proposed by Cairns et al. (1997). More detailed information about forest inventories and storm events occurring at each site is available in Supplementary Material I (Table S1).

Litter input from fine roots (here defined as roots of 5 mm in diameter), especially the finest ones with a diameter 2 mm, can significantly contribute to carbon sequestration in soils (Brunner et al., 2013; Kögel-Knabner et al., 2002; Berg and McClaugherty, 2008). Fine root litter was assumed to be proportional to that of foliage, which was measured on the RENECOFOR sites (see the following paragraph). Jonard et al. (2017) suggested using the generic equation published by Raich and Nadelhoffer (1989) and, simultaneously, adopting the hypothesis that fine root litter production represents about one third of the carbon allocated to roots (Raich and Nadelhoffer, 1989):

$$I_{fine\ root} = 0.333 \times 1.92 \times (100 \times I_{foliage}) + 130 \times 0.01 \qquad \text{(Eq. 4)}$$

Where, $I_{fine\ root}$ and $I_{foliage}$ are the litter input of fine roots and foliage, respectively (in tC ha$^{-1}$ year$^{-1}$).

The relationship between fine-root and foliage litter inputs can be highly variable depending on tree species, stand characteristics and climate, and the generic equation may not reliably represent such variability. To counter this, we carried out a sensitivity analysis to investigate the response of the model fit to the choice of fine-root:foliage ratio varying from 0.1 to 4.0 (see Sect. 2.6 and 3.2). Yet, when applying Raich and Nadelhoffer (1989)'s equation (Eq. (4) over all the RENECOFOR sites, we found that fine root:foliage ratios had a median of 1.0 and a mean of 1.0 – 1.1 for both coniferous and broadleaved sites (Fig. S2). Hence, we chose to use the 1.0 ratio over all the RENECOFOR sites to present the outcomes of model fit and residual analyses from the simulations (see Sect. 3.3). This facilitates our evaluation of site factors (e.g. dominant tree functional type, climatic and soil features) without adding a source of variability introduced by fine-root:foliage ratio.

**2.4 Litter carbon quality**

In the RENECOFOR network, there are no measured data for litter carbon quality, defined as the relative amount of litter carbon belonging to four different carbon pools (A, W, E and N). Therefore, we carried out a meta-analysis of the data collected in the literature where authors used chemical fractioning procedures or near-infrared spectroscopy (NIRS) techniques to measure litter carbon quality. The data was restricted to non-tropical areas. Chemical data on litter composed of coarse tree organs (e.g. stems, coarse branches) are relatively scarce, so we used

the tree stemwood data compiled in Pettersen (1984), Rowell et al., (2005) and Rowell
(2012). ~~Assembly of these works~~These three studies covers a wide range of temperate tree
species from North America, Japan and Russia~~, but no data are available for Europe~~. Data on
foliage and root litter carbon quality were ~~manually searched~~taken either from ~~either~~
networks~~, e.g.~~ such as CIDET (Trofymow et al., 1998) and LIDET
(http://andrewsforest.oregonstate.edu/research/intersite/lidet.htm) or from independent studies
in the northern hemisphere~~, including Europe~~. The database we used for ~~the~~ our meta-analysis
is available in Supplementary Material II. Root diameter or branching order can play a
significant role in modifying the composition of ~~the~~ soil chemical compounds (Fahay et al.,
1988; Tingey et al., 2003; Guo et al., 2004). All the measurements included in ~~the~~ our meta-
analysis on roots refer to fine roots (diameter < 5.0 mm), although in several studies, e.g.
Aber et al. (1990), Aulen et al. (2011) and Stump and Binkley (1993), root size was not
clearly indicated. Yet, we still included the data from ~~these above~~the latter studies, as
~~available root~~ data are less abundant for roots than foliage. The ~~collected~~ coarse-~~-root~~s data in
the literature were too few for a meaningful meta-analysis; we therefore used ~~and thus~~
stemwood values ~~for stemwood were used~~ instead.
We then used the resulting litter carbon quality database to describe the quality of litter input
~~of~~ at each site ~~of~~ in our study. ~~Partitioning of~~We portioned the litter input~~s~~ into biochemical
classes ~~respects~~ in the following order of priority: (i) values for the target species, when
available in the database; (ii) mean values of the species from the same genus, if data for the
target species ~~are~~ were absent~~;~~, and (iii) mean values of the species from the same tree
functional type (conifers versus broadleaves), if no data ~~are~~ were available at ~~n~~either species
~~n~~or genus level for ~~the~~a target species (see Table 1).
**2.5 Initialization of soil carbon quantity and quality**
To initialize Yasso07, both the quantity and the quality of the soil carbon are required. Here,
the initial carbon stock quantity was fixed ~~to~~ as the soil carbon stock measured ~~at~~ during the
first RENECOFOR soil carbon assessment ~~of the RENECOFOR~~ (i.e. ~~a~~ model input).
Measurement uncertainties of the soil carbon stock were not considered ~~as~~ to be a source of
stochastic effect when Yasso07 was fed, as we were more interested in the output
uncertainties related to the model per se (i.e., the choice of ~~model~~ the model's parameter set)
and in carbon quality settings in the model initialization (see below).
~~The~~ Soil ~~c~~Carbon quality, defined as the ~~the~~ relative amount of soil carbon in pools (A, W, E,
N and H) in relation to their sum, can be initialized ~~following two approaches~~in two ways:

with a complete steady-state assumption or with a partial, or transient, steady-state assumption. The classical approach is based on the assumption that carbon quality at initial state is identical to that at the complete steady-state, which can be calculated ~~using~~ with the analytical matrix inversion approach based on Eq. 1a. ~~At~~ For steady-state carbon stock~~s~~ ($t = t_s$), carbon gain is equal to carbon loss. Setting $\dot{X}\ t_S = 0$, (Eq. 1a) becomes:

$$A_pK\ c\ X\ t_S + I\ t_S = 0 \qquad\qquad (Eq.\ 5)$$

Solving (Eq. 5), we obtain~~ed~~ a steady-state carbon stock at time $t_s$: $X\ t_S$ :

$$X\ t_S = -\left(A_P K(c)\right)^{-1} I\ t_S \qquad\qquad (Eq.\ 6)$$

~~w~~Where $I\ t_S$ is a constant vector.

The estimated carbon quality in steady-state carbon stock $X\ t_S$ to the depth of 1.0 m (also noted as $C_{steady\text{-}state}$, in tC ha$^{-1}$) was then applied to the observed carbon stock to split it in~~to~~ various carbon pools.

The complete steady-state assumption is commonly used in the literature despite ~~high~~ considerable controversy ~~as such~~since the assumption does not ~~consider~~ take into account the difference in stabilization among the~~se~~ various pools (Elliot et al., 1996; Foereid et al., 2012). Soil carbon pools (especially those at sites that ~~underwent~~ have undergone disturbances in recent centuries) may not ~~be in a~~have achieved a complete steady-state, but may still be in a transient or partial steady-state. In such states, the slow-cycling pools can ~~be~~ still be accumulating carbon, while the relatively rapid-cycling pools ~~are able to~~have already recover~~ed~~ ~~until~~ a dynamic equilibrium (Wutzler and Reichstein, 2007). In this study, we equally adopted the partial steady-state assumption to mimic such a circumstance. More precisely, we assumed that the rapid-cycling pools such as A, W and E were at steady-state at the first soil survey, while the slow-cycling N and H pools might not yet have reached the steady-state ~~yet~~. Accordingly, ~~while directly considering~~we directly considered the steady-state CS obtained from matrix inversion ~~as~~for A, W and E, but we revised amounts for the N and H pools ~~amounts~~ by calculating the difference ~~with~~between estimated and~~the~~ observed CS ~~until~~to a depth of 1.0 m. In most cases, the sum of steady-state A, W, E and N was lower than the observed CS; the revised H was then equal to the difference between the latter and the former. Very occasionally, the sum of steady-state A, W, E and N could be greater than the observed CS; the revised N was then calculated ~~by~~as the difference between observed carbon stock~~s~~ and pool H was forced to zero. The new carbon quality, ~~which~~correspond~~ing~~s to the proportions among the steady-state A, W and E pools and the revised N and H pools, ~~will be~~was used to split the observed CS into five pools in real simulations.

## 2.6 Sensitivity analyses on the impact of initial soil and litter settings on model output

It is important to gain a general idea of the magnitude of the impact on model output and fit of our choices ~~of~~ for initial soil and litter settings in the process of model initialization ~~on model output and fit~~. To this end, we carried out a sensitivity analysis to assess how assumptions on carbon quality (complete steady-state versus partial steady-state) and carbon quantity as a function of soil depth (observed CS ~~until~~ to a depth of 1.0 m versus observed CS ~~until~~ to 0.4 m) and of fine-~~-~~root: foliage ratios (from 0.1 to 4.0) affected model predictions. Model fit is expressed via the comparison between simulated and observed annual changes in carbon stocks ~~changes~~ in the soil (ACC).

~~Besides~~In addition, we conducted another sensitivity analysis to fully explore the effects of all the theoretical~~ly possible~~ initial soil carbon quality distributions and that of simulated duration ~~simulation length~~ on model outputs~~, we conducted another sensitivity analysis~~. ~~For this, we~~We created a virtual site where ~~the~~ climatic conditions and litter input were constant and equal to the average values of all the RENECOFOR sites. By fixing its initial soil carbon stock to 100 tC ha$^{-1}$, we permuted the initial percentage of the soil carbon pools, with ~~the following constraint: the~~ ~~minim~~um~~al~~ and maximum percentages fixed ~~are~~ at 5% and 80%, respectively. We used four levels of ~~simulation length~~simulated duration (1~~;~~, 10~~;~~, 100~~;~~, 1,~~-~~000 and 10,~~-~~000 years) for each combination of soil carbon quality distribution. Based on averaged soil and litter carbon data ~~of~~from the RENECOFOR sites, the simulations were carried out for both broadleaved and coniferous forest ~~stand cases~~types. Here, we present only the results ~~of~~for the virtual broadleaved stand~~s case were presented~~, as the results between conifers and broadleaves did not change much, especially ~~in~~over the long term.

## 2.7 Running Yasso07 and statistical analyses

We used the same FORTRAN code ~~of the~~as Yasso07 version 1.0.1 ~~used~~ in Didion et al. (2014) for all the model simulations. For each type of analysis (both RENECOFOR site-specific and sensitivity analyses), we conducted ten~~10~~ simulations. In each simulation, one parameter vector was randomly chosen from the 10,~~-~~000 parameter vectors.

For each site, we calculated changes in annual carbon stocks ~~changes~~ (ACC, in tC ha$^{-1}$ year$^{-1}$), i.e., the difference ~~of~~ in carbon stocks between the two national ~~inventories~~RENECOFOR assessments standardized by the temporal interval ($t_2 - t_1$) as follows:

$$ACC_{obs} = (CS_{obs,t2} - CS_{obs,t1})/(t_2 - t_1)$$
$$ACC_{sim} = (CS_{sim,t2} - CS_{obs,t1})/(t_2 - t_1)$$

(Eq. 7a and 7b)

where, $CS_{sim,t2}$, $CS_{obs,t2}$ and $CS_{obs,t1}$ are~~.~~, respectively, the simulated carbon stock ~~until~~ to a depth of 1.0 m at ~~the~~ year $t_2$, and the observed carbon stock at ~~the~~ year $t_2$ and $t_1$, ~~which are~~that is around ~~the year of~~ 1994 and 2010 depending on ~~each~~ the site~~, respectively~~.

To compute $ACC_{sim}$ (Eq. 7b), some previous studies used a simulated CS at the starting year instead of an observed one (e.g. Ortiz et al., 2013). In such a case, it is of primary importance to judge a "steady-state year" prior to the starting year ~~from~~ for which observed data are available. From the estimated steady-state year, a spin-up or real model simulation is then followed to obtain a simulated CS at the starting year. In our simulations, the observed soil carbon stock at $t_1$ ~~was~~ served as ~~the~~a model input to set initial soil quantity and to calculate ACC (Eq. 7b). This allows avoiding such a judgement on steady-state year, which can be sometimes subjective. This ~~also~~ allow~~ed~~s us to better focus~~ing~~ on the effect of initialized soil carbon quality, for which we ~~attempted~~ calculated both complete ~~or~~ and partial steady-state assumptions (see Sect. 2.5).

Two reasons support our ~~general preference of comparing~~choice to compare $ACC_{sim}$ with $ACC_{obs}$ instead of comparing $CS_{sim,t2}$ with $CS_{obs,t2}$. First, the parameter sets of Yasso07 were calibrated for a soil depth of 1.0 m, while carbon stock data from the two RENECOFOR assessments ~~at the RENECOFOR sites~~ were only available ~~until~~ to 0.4 m (because ~~the~~ no data ~~of~~ from 0.4 - 1.0 m in depth ~~from~~ were available from the ~~2nd~~ second assessment ~~are unavailable~~). It is ~~thus~~ therefore reasonable to ~~speculate~~ assume that the observed carbon stock data are not comparable with Yasso07 estimates. However, focusing on carbon changes instead of carbon stocks may largely erase this bias~~.,~~ ~~because~~ Indeed, previous studies have evidenced that carbon dynamics are much less active at deep soil layers than at superficial layers (Jandl et al., 2014; Balesdent et al., 2018). Second, ACC indicates if a site is gaining or losing soil carbon and this information is sometimes more important than the site's carbon stock value. Using a metric standardized ~~metric~~ (by year~~)~~ such as ACC can also facilitate comparing result~~s~~ ~~comparison for~~in future studies. The only exception was for our sensitivity analysis on the effect of initial soil carbon quality (Sect. 2.6), where we chose $CS_{sim,t2}$ instead of $ACC_{sim}$, since the initial soil carbon stock was fixed at 100 tC ha$^{-1}$. Despite ~~the~~ our primary focus on ACC, we ~~additionally~~ also compared the simulated steady-state carbon stock ($CS_{steady-state}$, in tC ha$^{-1}$)~~, which was~~ obtained from the initialization procedure (see Sect. 2.5)~~,~~ with the $CS_{obs,t1}$ down to 1 m ~~soil~~ in depth~~;~~ this was ~~in order~~ to check if Yasso07~~'s~~ was able

~~to~~ predict~~ed~~ stocks to 1.0 m depth that indeed reach~~ed~~ the level of observed stocks (see Fig. S4).

~~In order to~~To test the performance of Yasso07 in estimating changes in soil carbon ~~changes~~ at the RENECOFOR sites, we used an analysis of variance (ANOVA) to analyze ~~analyzed~~ the residuals of the changes in carbon ~~changes~~, here defined as the difference between the simulated and observed values.~~, using analysis of variance (ANOVA).~~ The following environmental and biological factors were tested: site geographical location (latitude, longitude, and altitude), climatic conditions (temperature and precipitation), soil type~~s~~, tree functional type and tree species. Before each ANOVA, we tested the normality of the data ~~using~~ with a Shapiro – Wilk test. For the sensitivity analyses, we performed loess regressions (Fox and Weisberg, 2011) to characterize the variations of soil carbon stocks as a function of the initial soil carbon stock settings and ~~simulation length~~simulated duration (1 – 10,000 years). Statistical analyses were performed ~~using~~ with R 2.13.0 (R Core Team, 2013).

**3 Results**
**3.1 Litter carbon quality of northern temperate tree species**
Our meta-analysis (Fig. 2) showed that the litter carbon quality, i.e. carbon composition, of
northern temperate tree species significantly differed between tree organs. For woody litter
(stem data alone) the A carbon pool reached up to 80% of
the total carbon pool; the sum of the A and N carbon pools corresponded to at least 75%
and, in most cases, was greater than 90%. W
and E accounted for only small percentages of the carbon composition (Fig. 2a). Nevertheless,
this dominance of A and N over W and E was much less pronounced in foliage and root litter
types (Figs. 2b and 2c). Generally, the different tree organs can be ranked according to the
sum of the proportions of A and N as follows: woody debris (>90%) > roots (70 – 80%) >
foliage (60 – 70%, Fig. 2d).
The effect of tree functional type on litter carbon quality strongly interacted with that of tree
organ. For wood, broadleaves and conifers had clearly shifted point clouds for the
relationship between A and N carbon pools: there was a greater proportion of A, and a
lower proportion of N in broadleaves than in conifers. In foliage and root
litter, the effect of tree functional type on the proportions of A and W was less pronounced
than for wood. The main difference between broadleaves and conifers occurred in N rather
than in A (Fig. 2d). Broadleaved litter had a smaller N proportion than did
coniferous litter regardless of tree organ (Fig. 2d). The proportions of A and N relative to
those of E and W were quite stable between broadleaves and conifers regardless of tree organ
(Fig. 2d).
**3.2 Sensitivity analyses on the impact of initial soil and litter settings on model output**
Fig. S3 showed the impact of different settings of litter and carbon quantity and quality on
model fit for the RENECOFOR sites. For soil carbon quality, the partial steady-state
assumption (Fig. S3c and S3d) achieved significantly better model fits (with lower model
root-mean-square-error) than did the complete steady-state assumption (Fig. S3a and S3b).
Next, we found that model fits were better when observed CS to 0.4 m in
depth was used as the initial carbon quantity than when CS to 1.0 m in depth
was used (Fig. S3a and S3c). Nevertheless, it remained more advantageous to
use the CS to 1.0 m observed during the first assessment as the model input
is more advantageous, because Yasso07 is calibrated to predicts CS down to 1.0 m in depth
due to its used datasets for model calibration (Rantaraki et al., 2012).
Different choices of fine-root:foliage ratio for fine root litter input also significantly
influenced Yasso07's performance in predicting changes in soil C changes (Fig. S3). Ratios
of 0.1 – 0.8 for broadleaves and 1.8 – 3.0 for conifers achieved the best fits between simulated
and observed changes in soil CS changes according to different scenarios (Fig. S3). Using a
constant value of 1.0 for both broadleaved and coniferous sites seems to be an acceptable
compromise between both the two tree functional types, although even though such athe
choice is not optimal for each single functional type taken individually.
Based onAs a result of the above diagnoses, we only show only fit and residual analysis
results based onfor the simulations with based on the partial steady-state assumption, the
observed CS until to 1.0 m and fine-root: foliage ratio of 1.0 (see Fig. S3d) were shown in the
and Sect.3.3).
Fig. S4 visualized all the theoretically possible final carbon stocks by varying initial carbon
stocks and simulation lengthsimulated duration (from 1 to 10,000 years). IThe initial soil
carbon quality had a pronounced impact on the final soil organic carbon stocks at the annual
and decennial scales. For example, when the initial proportion of the A pool increased from 0
to 80%, the final proportion of A could increase by +30 to +40 tC ha$^{-1}$ (Fig. S4a) and the final
total carbon stock could decrease by approximatelye.a. -20 to -30 tC ha$^{-1}$(Fig. S4u) at annual
and decennial scales. When simulations were performed over for a millennium timescale, the
initial soil carbon quality did not no longer impact the final soil carbon quality anymore. In
other words, the same final soil carbon quality was obtained regardless what of the initial soil
quality was (Fig. S4).

### 3.3 Simulated versus observed carbon data

Using only mean litter input, the theoretical carbon stocks ($CS_{steady-state}$) simulated from the
initialization method and the observed $CS_{obs,t1}$ to 1 m in depth shared the same order of
magnitude and were even quite comparable (Fig. S5). However, the carbon stocks were
overestimated for most coniferous stands, and underestimated for broadleaved stands (Fig.
S5).
When simulated annual changes in carbon stock changes (ACC) were plotted against
observed ones, the point clouds were distributed around the 1:1 diagonal line despite fairly
high dispersion (Fig. 3). The correlation between predicted and measured ACC remained
weak ($R^2 < 0.1$). The mean observed and simulated annual changes in carbon stocks changes

(ACC) ~~of~~ for all sites ~~are~~ were, respectively, +0.34 ± 0.06 tC ha$^{-1}$ year$^{-1}$ (+0.20 ± 0.06 tC ha$^{-1}$ year$^{-1}$ for broadleaved stands and +0.48 ± 0.10 tC ha$^{-1}$ year$^{-1}$ for coniferous stands) and +0.00± 0.07 tC ha$^{-1}$ year$^{-1}$ (+0.28 ± 0.09 tC ha$^{-1}$ year$^{-1}$ for broadleaved stands and -0.28 ± 0.11 tC ha$^{-1}$ year$^{-1}$ for coniferous stands)~~, respectively~~. Thirty-two ~~32~~% of the broadleaved stands and 39% of the coniferous stands showed significant differences between observed and simulated ACC (Fig. 3a). In only approximately~~c.a.~~ 17% of the sites, ACC ~~were~~ was significantly different from 0 for both simulated and observed results (i.e. ~~the~~ case 3 in Fig. 3b). Here, t~~T~~here ~~is~~ was a significant effect of ~~the~~ tree functional type on the observed and simulated values. The model tended to overestimate ACC in broadleaved stands but to underestimate ACC in coniferous stands. ~~The quantity~~ Approximately two thirds of all the ~~of~~ sites ~~in which~~showed estimate~~d~~s and observed changes in carbon stock~~s~~ ~~changes share~~ of the same ~~tendency~~ trend (i.e. data points in ~~the~~ zone~~s~~ I, III, IV~~, III~~ and VI, Fig. 3)~~. was~~ ~~approximately two thirds of the~~ while ~~total sites.~~ approximately~~c.a.~~ one third of the sites ~~are~~ were in the remaining zones (II~~,~~ and V) where the predicted ~~tendency~~ trend was contrary to the observed ~~tendency~~trend. From the residual distribution, we ~~could~~ also ~~find~~ found that the model ~~fit where~~with carbon quality was set ~~by~~ with the partial steady-state assumption (Fig. 3) ~~was~~ had better fit than ~~that~~ the model set ~~by~~ with the complete steady-state assumption (Fig. S6).

The simulated changes in carbon stock~~s~~ ~~changes~~ exhibited a negative linear relationship with the initial soil carbon stock (Fig. 4b), whereas this ~~tendency~~ trend was not ~~observed~~ found for the observed changes in carbon stock ~~changes~~ (Fig. 4a). Storm damage and soil type could not ~~provide~~ clearly ~~tendencies in explaining~~explain these trends in the residuals. ~~Only~~ For coniferous stands only, the residuals showed significant~~ly~~ differences among the three major types of soil ($n$ of sites >5): cambisol > luvisol > podzol (Fig. S7). ~~Tree ages in~~The coniferous stands tend~~ed~~ to be ~~smaller~~ younger than ~~those in~~the broadleaved stands. ~~When considering~~ Neither tree age nor the interaction between tree age and~~both~~ tree functional type~~s and tree~~ ~~ages, neither the latter nor their interaction~~ had a~~ny~~ significant effect on residuals. ~~With~~ For all the sites together, the residuals bec~~a~~ome higher with increasing latitude, indicating that simulated ACC was more overestimated in northern zones (ANCOVA, F = 11.2, $P<0.001$). This pattern was particularly strong for broadleaved stands (Fig. S8a). ~~Yet, this tendency~~No similar trend was ~~not clear~~found for coniferous stands (Fig. S8e). Both residual signs were generally present for all of the main species (Fig. S8b, S8c, S8d, S8f, S8g and S8h). Broadleaved and coniferous stands differed in their responses to environmental factors: for coniferous stands, ~~both~~ neither temperature ~~and~~ nor precipitation had ~~little~~ much effect on

residuals, whilst for broadleaves, precipitation was negatively correlated with residuals
(ANCOVA, F = 10.8, *P*<0.001).
Regarding soil physical and chemical properties, total soil nitrogen stocks soil were
significantly correlated with residuals for both broadleaved and coniferous stands (Fig. 5).
Then, soilSoil texture (proportions of clay and sand) and exchangeable magnesium and
potassium were significantly correlated with residuals only for broadleaved stands (Figs. 5
and S9 Table S2). The remaining tested variables, such as exchangeable aluminum and
calcium, pH, total phosphorus and carbon:nitrogen ratio, had showed no relationships with the
residuals (Table S2).

1. **4 Discussion**

2. **4.1 Agreement between simulated and observed annual changes in soil carbon stock ~~changes~~**

3. Testing widely popularized soil carbon models ~~using~~ on a large dataset is highly meaningful

4. work that enables researchers not only ~~assessing~~ to assess the model's predictive ability over

5. various climatic and ecosystem types, but also provid~~es~~ing lessons and implications for future

6. modelling work. Here, ~~based on~~compared with ~~the~~ observed carbon stock data to 1.0 m in soil

7. depth from the RENECOFOR network, we found that the simulated ~~and observed carbon~~

8. stocks ($CS_{steady-state}$ versus $CS_{obs, \, t1}$) to 1.0 m showed the same order of magnitude, and

9. validat~~ed~~ing Yasso07's ~~good capability~~ability to predict average carbon stock~~s~~ ~~in average~~ at

10. the scale of the French territory. This ~~good~~ solid performance at the national scale ~~is~~

11. ~~consistent with~~ supports Yasso's aim ~~for generality~~ to be generalizable and ~~supported by~~ is

12. consistent with previous studies (see Ortiz et al. 2013; Lehtonen et al. 2016; Hernández et al.

13. 2017). Nevertheless, ~~the~~ observed CS ~~until~~ to 1.0 m in depth at *t1* already exceeded the

14. ~~already~~ $CS_{steady-state}$ for most coniferous stands (Fig. 5S), suggesting, to some extent, ~~some~~

15. ~~inadaptability of~~that the model parameters were not adapted to the RENECOFOR dataset.

16. Such inadaptability may simply be due to ~~the~~ setting ~~of~~ an over~~ly~~ high decomposition rate ~~of~~

17. for the slow carbon pools in the model~~.~~ As~~Or, as~~ the coniferous stands ~~are~~were on average

18. younger ~~and were~~being afforested more recently than the broadleaved stands (Jonard et al.,

19. 2017), the model ~~does~~may also not have been able to account for ~~such~~ historic land use

20. change~~s~~ ~~history to calculate~~when the SOC stock was calculated at the steady state. Fig. S5

21. ~~also showed~~shows that for most broadleaved stands, observed stocks ~~are~~were lower than their

22. $CS_{steady-state}$, possibly ~~forming the evidence that~~indicating that steady-state equilibrium ~~may~~

23. had ~~have~~ not yet been reached at these sites.

24. ~~Then, based~~Furthermore, compared with ~~on~~ the average observed annual changes in soil

25. carbon stock~~s~~ ~~changes~~ (ACC) ~~with average~~over the ~~15-year interval between the two

26. ~~inventories~~assessments, we found that the simulated ACC were significantly biased for more

27. than one third of the French RENECOFOR sites. Particularly, Yasso07 generally

28. overestimated the ACC at the broadleaved stands located in the north of France (Fig. S8a-d);

29. ~~and the~~this overestimation ~~can be~~was sometimes exacerbated ~~with~~by lower precipitation. On

30. the other hand, Yasso07 tended to underestimate the ACC in ~~our~~ the coniferous stands.

31. Nevertheless, we ~~would~~expect~~ed~~ ~~slightly better performance of~~Yasso07 to perform slightly

32. better in the coniferous stands than in the broadleaved ones, since the model's estimates have

33. shown good correspondence to measurements (of stocks and/or changes) in coniferous

forests, especially ~~the~~ Nordic boreal ~~ones~~ forests (e.g., Karhu et al., 2011; Ortiz et al., 2013). Probably due to the younger age of the coniferous stands in our study, the observed ACC of the coniferous stands were greater than those of the broadleaved stands (Fig. 3, Jonard et al;, 2017). Again, Yasso07 was unable to reproduce this observed effect of tree functional type on ACC, as ~~it~~ the model ~~lacks consideration of~~does not take into account changes in land_use ~~change~~ history, ~~i.e., the same reason with~~as for the case of steady-state carbon stocks mentioned above.

Except for tree functional type and geographical location (e.g. latitude, which is correlated with climatic variables), qualitative ecological variables that are assumed ~~as~~to be key factors influencing carbon sequestration processes~~,~~ (e.g. soil type ~~-~~ (except for coniferous stands~~)~~, storm damage and stand age range), showed limited ~~tendencies in explaining~~ability to explain residuals. Note that ~~those~~these factors were not fully crossed ~~in~~for the 101 sites, ~~rendering~~ making it difficult to test~~ing~~ each single factor.

~~The s~~Simulated ACC showed a strongly negative correlation with the observed initial soil carbon stocks ($CS_{obs,t1}$)~~-~~, with an overestimation of ACC at sites ~~of~~with lower CS $_{obs,t1}$ and an underestimation at sites ~~of~~with higher CS $_{obs,t1}$ (Figs. 4 and S9). ~~Such phenomenon~~This ~~can be logically explained by~~ is logical in view of the model's inherent mechanism. With increasing initial carbon stock, there is an increase in the quantity of ~~those~~the easily decomposable compounds in the soil, i.e. A, W and E, ~~in soil,~~ which triggers a more substantial mass loss at a decennial scale. However, the ~~observed~~data on observed changes in carbon stocks ~~changes~~ did not support this trend.

Several quantitative soil physical and chemical properties showed clear correlations (especially for broadleaved stands) with ACC residuals (Fig. 5). Also, in the principle component analyses (Fig. S9), the arrows ~~standing for~~representing soil variables are slightly closer to the pivoting axis of "initial carbon stock – ACC residuals" than those representing ~~standing for~~climatic and geographic variables, notably for broadleaved stands. These results ~~suggest~~highlight the~~a~~ potential interest of incorporating soil properties into new versions of the Yasso model ~~family~~, ~~in~~which currently lacks, or only implicitly incorporates, soil parameters ~~are lacking or only implicitly incorporated~~. Indeed, there ~~are numerous~~is considerable evidence~~s~~ that soil physical and chemical properties can greatly ~~govern~~influence soil carbon dynamics and ~~stock~~storage capacity (Beare et al., 2014; Dignac et al., 2017; Rasmussen et al., 2018).

~~The limitations of the model at the site-scale are not surprising as the model was developed primarily for primarily large scale applications.~~ Despite Yasso07's significant prediction bias

at a number of sites, it is unreasonable to simply attribute the bias to the model *per se*, ~~as~~ since multiple uncertainties affecting the quality of the model's input data can be identified (see Sects. 4.2 – 4.3). These uncertainties can occur not only with Yasso07, but also with other prevailing models ~~one may choose~~, highlighting large knowledge gaps in ecology and soil carbon modelling.

**4.2 Setting soil carbon quality for model initialization: a recurrent challenge in soil carbon modelling**

~~GA~~ great uncertainty is associated with ~~the~~ model initialization ~~of~~ in terms of soil carbon quality, as it ~~was~~ is usually estimated, not measured, ~~but usually estimated,~~ for example, ~~by~~ through matrix inversion with the assumption that the litter input has been the same for decades. Compared to measuring total soil carbon stocks, measuring soil carbon quality is much more labor-intensive and time-consuming. Moreover, soil carbon quality data ~~of soil carbon quality~~ from different sources ~~are~~ may be partly or totally incompatible due to the use of different chemical pools or fractionation protocols ~~of fractionation~~ (Blair et al., 1995). Therefore, measured data ~~of~~ for soil carbon quality are generally lacking at worldwide scale. ~~Such~~ This lack of information is a recurrent issue for soil carbon dynamics modeling (see Elliot et al. (1996), who has discussed the issue of "Measuring the modelable"). Many prevailing soil carbon models require setting carbon quality ~~besides~~ in addition to carbon quantity, e.g., Romul (Chertov et al., 2001), RothC (Coleman and Jenkinson, 1996), CENTURY versions Parton et al., 1987; Metherell et al., 1993, CBM-CFS3 (Kurz et al., 2009). ~~Inappropriate S~~setting ~~of~~ carbon quality in models inappropriately may greatly change carbon stock prediction~~s~~ (Wutzler and Reichstein, 2007; Carvalhais et al., 2008; 2010).

In ~~the present~~this study, soil carbon quality data were unavailable at the French RENECOFOR sites. We therefore tested both complete and partial steady-state assumptions for setting the initial carbon quality. Compared to the complete steady-state assumption, the partial steady-state assumption ~~allows that~~made it possible to account for slow cycling pools, which could ~~can~~ still be ~~still~~ accumulating carbon, ~~while~~ and fast cycling pools ~~are~~ in equilibrium (Wutzler and Reichstein, 2007). We ~~In this study, w~~e did not use the ~~exact~~ precise method ~~to estimate initial carbon quality as~~ proposed ~~in~~ by Wutzler and Reichstein (2007) to estimate initial carbon quality due to ~~the~~ a lack of information ~~for setting the modified(??)~~necessary for the decomposition-accumulation dynamics of the H pool. ~~Nevertheless~~Instead, ~~following~~ while we followed the same ~~idea of~~ partial steady-state assumption, we revised the proportions of the N and H pools ~~by~~ and assumed~~assuming~~ that

the A, W and E pools ~~are~~ were in equilibrium and equal to the simulated ~~steady-state~~ ~~ones~~values. ~~and that~~We also assumed that the sum of all pools at *t1* ~~is~~ was equal to the observed stock. We found that our partial steady-state assumption gave rise to generally better model fits than did the complete ~~one~~ steady-state assumption (Fig. S3; see also Figs. 3 and S6), ~~hinting~~ indicating its good suitability to the RENECOFOR sites. When plotting $CS_{stead-state}$ against $CS_{obs}$ (Fig. S5), we ~~visualized the~~found a discrepancy:~~that,~~ while the $CS_{obs}$ of most of the broadleaved stands were smaller than $CS_{stead-state}$, the $CS_{obs}$ of most of the coniferous stands were greater than $CS_{stead-state}$. ~~Such a~~This discrepancy was ~~then~~ brought into ACC fit when the complete steady-state assumption was adopted (Fig. S6). Nevertheless, the partial steady-~~steate~~state assumption can, to some extent, mitigate such discrepancies.~~:~~ ~~F~~for broadleaved stands, the revised proportions of the A+W+E pools became higher than those at the complete steady-state (Fig. 6:~~;~~ with 70% of stands above the ~~the~~ steady-state ~~strip~~line), thus reducing the model's overestimation of ACC.~~;~~ ~~F~~for coniferous sites, the proportions of the A+W+E pools ~~are~~ were often compressed (Fig. 6:~~;~~ with ~~<~~>50% of the stands ~~below~~ above the steady-state strip), thus reducing the model's underestimation of ACC at the steady-state. For future work, it would definitely be ~~definitely~~ worthwhile to ~~have~~ compare both assumptions ~~compared using~~for several prevailing carbon models (e.g., Yasso07, RothC, Century etc.), as studies comparing initialization assumptions still remain ~~scanty~~ scarce compared to those on model comparisons.

In order to gain a global overview ~~on~~ of Yasso07's sensitivity to initial soil carbon quality, ~~here~~ we ~~equally~~ also conducted a sensitivity analysis that computed the final soil carbon stocks ~~using~~for all ~~the~~ possible ~~combinations of the~~chemical pool composition~~s of chemical~~ ~~pools~~. This sensitivity analysis confirmed the high influence of initial soil carbon quality on soil carbon stock estimates (Fig. S4), notably at short temporal scales (i.e., yearly and decennial). This result is in line with ~~the~~ previous carbon stock modelling studies (Parton et al., 1993; Kelly et al., 1997; Smith et al., 2009; Foereid et al., 2012), confirming that ~~it~~ initialization is a crucial step for all ~~of the~~chemical-pool-based carbon models. ~~Besides this~~ ~~consensus, our~~Our sensitivity analysis further showed that ~~such~~ the effect of initial carbon stock composition ~~carbon stocks will~~ would gradually vanish with increasing length of ~~simulation~~ time, ~~and~~ especially ~~when the length is up to~~ in the case of several centuries or millenni~~aums~~. Our analysis provides new insights on the sensitivity of ~~model estimated~~ carbon stock~~s~~ estimates to the method and assumptions used in model initialization. ~~Such~~This analysis can be ~~transplanted~~ transposed to ~~the~~ other carbon models to test their theoretical

performance and robustness ~~of each model~~ at different temporal scales and ~~also,~~ to compare models.

Finally, ~~solely~~ testing different initialization assumptions ~~or~~ and performing sensitivity analys~~e~~is ~~does not allow radically solving~~are not enough to solve the predicti~~bility~~on issue~~s~~ related to uncertainties ~~of~~ in soil carbon quality. Based on ground truth data, Balesdent et al. (2018) showed that carbon age ~~shows strong patterns as a function of~~strongly reflects soil depth and ecosystem type. It appears highly necessary for future modelling work to capture better indicators ~~for~~ of carbon stabilization mechanisms~~,~~ in~~to~~ the ~~procedure of~~ model initialization procedure. ~~For this, it is to be noted that~~ Yasso07's particular model configuration, i.e. ~~the use of~~based on measurable chemical pools, may ~~open the possibility of using~~make it possible to use measured soil carbon quality data ~~of soil carbon quality~~ for model initialization instead of steady-state assumptions. Future measurements of radiocarbon age ~~on~~for soil ~~carbon~~ organic matter ~~radiocarbon age of~~at the RENECOFOR sites may offer an ideal opportunity to compare the impact of ~~the two sources of~~initial soil carbon quality on Yasso07's predictions.

**4.3 A precise estimation of root litter quantity helps improve Yasso07 prediction~~s~~**

An important source of uncertainty in the estimates of litter quantity at the RENECOFOR sites ~~was the~~ concerned fine root litter input. Many studies have revealed that fine roots ~~act as~~are a major source contributing to total litter quantity due to their fast turnover rates (Brunner et al., 2013; Kögel-Knabner et al., 2002; Berg and McClaugherty, 2008). In some forest ecosystems, the proportion of fine root litter is even comparable to that of foliage (Freschet et al., 2013; Xia et al. 2015). However, estimating fine root litter input~~s~~ is, again, a time-consuming and challenging task. ~~Due to~~For this reason, to our best knowledge, ~~so far rarely have~~probably no nation~~-~~al~~~ wide forest inventory projects have ever incorporated direct measurement~~s~~ of the dynamics of fine root litter input (~~i.e. the case of~~and this information is also lacking for the RENECOFOR network). Fine root turn-over~~s~~ ~~of~~for forest species ~~are variable~~varies depending on climate, tree species and management scenarios (Kögel-Knabner et al., 2002; Litton et al., 2003; Mokany et al., 2006), ~~rending the choice of~~and this makes choosing model input values highly subjective and difficult. By testing variable fine root:foliage ratios of litter input, we observed a significant shift in the ~~predicted~~ changes in carbon stock~~s~~ ~~changes~~ predicted by Yasso07 (Fig. S2). This finding not only highlights the importance of precisely ~~quantification of~~quantifying fine root litter input, but also suggests that broadleaves and conifers may have ~~separated quantification of~~a different fine-root litter

input ~~ratio~~ with regard to that of foliage, although ~~here~~ we chose the same ratio for both broadleaved and coniferous stands in this study. ~~We also~~also It should be noted that using one ratio per tree functional type (conifers versus broadleaves) ~~could~~ can only change the overall prediction baseline, ~~but~~ and cannot reduce ~~the~~ data dispersion. Consequently, it is of great interest to estimate fine-root litter input quantity at species level ~~on the basis of~~through direct measurements and then couple the specific data with Yasso07.

~~Another~~ Ppotentially important litter input~~s~~ may also come from the ~~understory~~ shrubby and herbaceous understory species, which ~~were not taken~~we did not take into account in this study due to data unavailability. The Hherb and shrub layer~~s~~ are typically not included in forest inventories ~~but~~ though they can contribute significantly to the annual litter production in forests (eg. de Wit et al. 2006, Gilliam 2007, Lehtonen et al. 2016). Muukkonen and Mäkipää (2006) estimated that the carbon input~~s~~ from herbaceous and shrub vegetation in Finnish forests ~~were~~ was ~~in the range of~~from 0.50 to 0.66 tC ha$^{-1}$ year$^{-1}$~~. Such value is apparently~~. This is quite high, as ~~it attains~~the value represents 12% - 23% of the mean total tree litter input~~s~~ ~~of~~ for all the RENECOFOR sites combined (Table 1). This is in line with ~~the~~ preliminary data from Etzold et al. (2014), who suggest~~ed~~ that understory vegetation contribute~~d~~ approximately~~e.a.~~ 12% (0.1 to 36.8%) to the total observed annual C turnover at six sites ~~of~~ in the Long-term Forest Ecosystem Research Programme LWF (Swiss part of the ICP Forests~~-~~Level II ~~plots~~network).

Finally, Yasso07's parameter set was calibrated ~~using~~ based on one of the richest litterbag datasets in the world in terms of number of observation~~s~~. ~~The~~ Sstate-of-the-art of soil carbon modeling ~~is based on the~~assumes that litter input and decomposition processes ~~as~~ are the driving forces in soil carbon accumulation. ~~Our knowledge on the importance of~~However, other important sources of biological carbon input exist, e.g. soil fauna and rhizodeposition; unfortunately, our ability ~~, as well as how~~ to take them into account in modelling processes ~~still~~ remains poor. ~~Accordingly,~~ Wwhether, and to which extent, the bias ~~of~~ found in our Yasso07 results is related to these alternative sources of biological carbon input is unknown.

**4.4 Suggestions for future modelling improvement~~s~~ ~~in the future~~**

First~~ of all~~, we found the Yasso07 model structure and algorithm ~~good~~solid, clear and simple to operate~~.~~ ~~and this goes along well~~in agreement with the positive remarks ~~toward Yasso and Yasso07~~ in the literature (Rantakari et al., 2012; Didion et al., 2014; Lu et al., 2015; Wu et al., 2015). Regarding its mass flow parameters, Fig. S1 only show~~ed~~ the mass flows that are statistically significant ~~for~~ in the case of ~~using~~ the Tuomi 2011 parameter set. Yasso07 keeps

all the theoretical mass flow possibilities in the $A_p$ matrix in (Eq. 1b). However, a mass flow parameter with a statistical significance does not signify that it is biologically meaningful. For ~~this~~ example, we can quote the flow N → A ~~of~~ in the model (Fig. S1), for which the modeler ~~had~~ assigned an astonishingly high percentage: $p_{N→A}$ = 83%. This quantity is disputable in ~~the angle~~light of soil biochemistry, because ~~as~~ lignin, ~~i.e.~~ the major component constituting the N carbon pool, is not likely ~~does not~~to ~~turn~~ pass into the A pool, but would instead probably condense with other nearby phenols, peptides or saccharides (Burns et al., 2013).

As a model ~~aiming at~~for predicting soil carbon dynamics, Yasso07 is still ~~highly~~ overly simple in the description of some soil variables that are known to strongly impact decomposition processes ~~in non organic soil~~. For example, ~~the effect of~~ soil mineralogy or aggregation ~~has not been considered in~~is yet to be accounted for in Yasso07 ~~yet~~. Indeed, the model ~~was~~ has often been applied on soils fairly rich in organic matter (e.g., Karhu et al., 2011), where the consideration of soil mineral properties was not particularly relevant, and where the authors' assumption that litter quantity is a good proxy for soil properties was reasonable. In addition, when Yasso, ~~i.e.,~~ Yasso07's prototype, ~~came up~~ was published in 2005 (Liski et al., 2005), information on mineral soil properties in the various forest soil horizons was not commonly available~~, but nowadays~~Nowadays, however, it is easier to obtain ~~it~~, although there is still ~~a lack of such~~not enough detailed data for consistent application across large regions or at the national scale (Didion et al., 2016).

**5 Conclusions**

We tested the performance of the Yasso07 soil carbon model ~~Yasso07 using the~~on decennial-scale French nation~~al~~-wide forest data ~~thank to~~collected through the RENECOFOR network~~,~~. ~~as well as~~We also compiled a meta-analysis database for litter carbon quality and carried out sensitivity analyses to characterize the effect of ~~inputs of~~initial litter input and soil carbon quality on the model's predict~~ion~~s. We showed that~~,~~ while the model's ~~predicts~~estimates of ~~the~~soil carbon stocks ~~until~~to 1.0 m ~~soil~~in depth and of changes in annual soil organic carbon ~~changes~~(ACC) stay~~ed~~ within the same order of magnitude ~~with~~as ~~the~~observed ~~ones~~values, the accordance between the observed and simulated ACC values at the site scale remained weak. There was a bias ~~of~~in the model~~'~~s predict~~ed trends~~ion for ~~the~~changes in carbon ~~change tendency~~stocks at more than one third of the French sites. ~~The performance of~~As we have shown for Yasso07, ~~as well as the other~~the performance of soil carbon models~~,~~ should be examined before their application ~~for~~to management guidelines and policy-making for forest ecosystems at any ~~study~~scales.

~~Such bias~~Biases can be attributed to multiple ~~reasons~~factors concerning model input, such as (i) ~~large~~uncertainty in ~~the measured~~the measurement data for soil carbon stocks and changes; (ii) a lack of information on initial soil carbon quality at the site level~~,~~ and (iii) a lack of information on below~~-~~ground litter production. These ~~reasons~~factors are valid for ~~the whole~~all state-of-the-art ~~of~~soil carbon modelling, regardless of the model that one uses. ~~For the latter two aspects, their importance was~~Our sensitivity analyses explicitly confirmed ~~by our sensitivity analyses~~the importance of factors (i) and (ii) above. ~~Appropriately s~~Setting soil carbon quality ~~should be~~is one of the most crucial step~~s~~ ~~influencing~~to guarantee the model's fit. ~~To set soil carbon quality, we~~We found that the partial ~~soil~~steady-state assumption gives rise to a significant~~ly~~ better model fit than does the complete steady-state assumption, when setting soil carbon quality. Some of the model's parameters governing the transfer among soil pools are statistically derived ~~but~~and not directly measured, and thus may poorly represent ~~the real~~actual biochemical decomposition processes~~of decomposition~~. Residual analysis also suggests a potentially important role of ~~soil~~physical and chemical soil properties in explaining the model's prediction ability.

~~These~~Our findings allow us to provide ~~a series of suggestions to~~modelers, users and policy makers with the following suggestions:

- ~~To Yasso07 modelers~~W, we suggest Yasso07 modelers keep~~ing~~ the current model structure, algorithm and parameter~~s natures~~, but incorporat~~eing~~ some more refined ~~some~~biochemical processes~~:~~, ~~including~~for example, that they (i) revis~~eing~~ certain

mass flows to achieve both statistically and biologically meaningful process~~es~~ (especially the N$\rightarrow$ A flow)~~:~~ (ii) refin~~eing~~ the decomposition process (i.e., the residence times between the A, W and E soil carbon pools)~~:~~ and possibly~~,~~ (iii) explicitly incorporat~~eing~~ eas~~ilyy~~-measured soil parameters to better represent biophysical and biochemical interactions in soil carbon cycling.

- ~~To Yasso07 users, we~~We suggest Yasso07 users work~~ing~~ in conjunction with modelers in order to better reduce the uncertainties in model initialization ~~of~~ for soil carbon stock~~s~~. We also suggest measuring forest carbon quality and quantity, and ~~also~~ belowground fine--root litter ~~data~~ to better feed the model.

- ~~To policy makers, we~~We suggest policy makers ~~keeping~~ remain prudent toward diagnose~~eis~~ ~~from~~ based on a single carbon model, especially when a long--term trend is predicted. Predictions from multiple models ~~served~~should be ~~as a cross-validation~~cross-validated ~~procedure are preconized~~ for both global and local ~~scales~~ areas.

~~Our~~ This study, involving decennial observation~~s~~ at sites ~~spreading at~~spread over a large spatial scale ~~that covers~~ and covering different ecosystems~~,~~ ~~can facilitate and~~ provide~~s~~ a good opportunit~~yies~~ ~~for~~ to facilitate future model calibration, improvement~~,~~ and re-assessment ~~of the model~~. Finally, ~~taking~~ with Yasso07 as an example, this work highlighted the bottleneck ~~of~~ in soil carbon modelling ~~due to lacking~~caused by the lack of knowledge or data on soil and litter carbon quality and on fine--root litter quantity, ~~rendering~~ which create high uncertainties for model ~~inputs~~initialization~~.~~, ~~and also demonstrated~~ Simultaneously, ~~this study~~we demonstrated methodologies ~~of~~ for testing ~~the~~ other soil carbon models via sensitivity analyses, ~~which~~ to better enable us to ~~better~~ understand the limits of the model and of the input data ~~input for~~ and to plan future improvements in soil organic carbon modelling. In this study, we used the ~~published~~ model structure and parameters ~~from~~ published in Tuomi et al. (2011a) without any modifications. ~~Upcoming~~ Further work ~~of~~ on sensitivity analyses incorporating modifications ~~of~~ in both the carbon quality and litter input settings ~~of carbon quality and litter inputs~~ and Yasso07's configuration and parameters ~~should be performed to~~is needed to ~~ultimately~~ confirm the reliability of the current diagnoses.

**Acknowledgement**
This study was funded by the French Agence de l'Environnement et de la Maîtrise de
l'Energie (ADEME, Contract ref. : 14-60-C0082). The UR1138 BEF and this study was
supported by a grant overseen by the French National Research Agency (ANR) as part of the
"Investissements d'Avenir" program (ANR-11-LABX-0002-01, Lab of Excellence ARBRE) –
QLSPIMS project. This study is an outcome of a project under taskentitled: "Input to
improveing the comparability in MRV across EU MS", within the LULUCF MRV project:
"Analysis of and proposals for enhancing Monitoring, Reporting and Verification (MRV) of
land use, land use change and forestry (LULUCF) in the EU" funded by the European
Commission. and fundingFunding for M. Didion was provided by the Swiss Federal Office
for the Environment. We thank several French colleagues Dr. I. Feix (ADEME), Dr. A.
Legout (INRA) and Dr. B. Guenet (CNRS) for their valuable comments to this work. We are
also grateful to Dr. A. Repo (FEI - SYKE) and Dr. E. Hilasvuori (FEI - SYKE) for their
explanations of Yasso07, and to Victoria Moore for her thorough review and suggestions for
improving the English language.

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

| Functional type | Species | Organ | Case | No. of obs. | | | | Mean (%) | | | | SD (%) | | | |
|---|---|---|---|---|---|---|---|---|---|---|---|---|---|---|---|
| | | | | A | W | E | N | A | W | E | N | A | W | E | N |
| Broadleaves | *Fagus sylvatica* L. | wood | 4 | 4 | 4 | 4 | 4 | 74.5 | 2.8 | 1.2 | 21.5 | 1.4 | 1 | 0.5 | 1.4 |
| | | leaf | 2 | 2 | 1 | 1 | 2 | 39.6 | 22.1 | 12.5 | 25.8 | 3.5 | NA | NA | 1.7 |
| | | root | 3 | 1 | 9 | 9 | 1 | 31.5 | 8.8 | 18.6 | 41.1 | NA | 1.2 | 1.2 | NA |
| | *Quercus petraea* (Matt.) Liebl. | wood | 4 | 19 | 19 | 19 | 19 | 67.5 | 6.1 | 3.5 | 22.9 | 4.9 | 2.3 | 1.7 | 2.6 |
| | | leaf | 4 | 12 | 12 | 12 | 12 | 40.8 | 16.3 | 14.2 | 28.7 | 3.5 | 4.7 | 9.3 | 7.1 |
| | | root | 5 | 15 | 9 | 9 | 15 | 34.9 | 7.6 | 16.2 | 41.3 | 8.0 | 1.1 | 1.1 | 10.4 |
| | *Quercus robur* L. | wood | 4 | 19 | 19 | 19 | 19 | 67.5 | 6.1 | 3.5 | 22.9 | 4.9 | 2.3 | 1.7 | 2.6 |
| | | leaf | 2 | 1 | 12 | 12 | 1 | 37.7 | 21.6 | 17.3 | 23.4 | NA | 7.3 | 7.3 | NA |
| | | root | 3 | 1 | 9 | 9 | 1 | 28.6 | 11.1 | 23.4 | 36.9 | NA | 1.5 | 1.5 | NA |
| Conifers | *Abies alba* Mill. | wood | 4 | 14 | 14 | 14 | 14 | 66.7 | 2.7 | 2.4 | 28.2 | 1.9 | 1.3 | 0.8 | 1.3 |
| | | leaf | 2 | 1 | 6 | 6 | 1 | 32.4 | 26.4 | 10.7 | 30.5 | NA | 1.4 | 1.4 | NA |
| | | root | 3 | 1 | 13 | 13 | 1 | 25.3 | 19.1 | 21.5 | 34.1 | NA | 6.2 | 6.2 | NA |
| | *Larix deciduas* Mill. | wood | 4 | 6 | 6 | 6 | 6 | 65.3 | 5.9 | 1.9 | 26.9 | 3.2 | 2.4 | 0.9 | 1.5 |
| | | leaf | 2 | 2 | 4 | 4 | 2 | 33.3 | 30.2 | 10.1 | 26.4 | 2.5 | 1.6 | 1.6 | 7.7 |
| | | root | 3 | 1 | 13 | 13 | 1 | 32.5 | 16.2 | 18.2 | 33.1 | NA | 5.2 | 5.2 | NA |
| | *Picea abies* (L.) H. Karst | wood | 1 | 1 | 1 | 1 | 1 | 69.5 | 1.9 | 1.0 | 27.6 | NA | NA | NA | NA |
| | | leaf | 2 | 1 | 6 | 6 | 1 | 37.0 | 29.5 | 12.0 | 21.5 | NA | 2.2 | 2.2 | NA |
| | | root | 3 | 3 | 13 | 13 | 3 | 36.6 | 14.8 | 16.6 | 32.0 | 7.8 | 4.8 | 4.8 | 2 |
| | *Pseudotsuga menziesii* (Mirb.) Franco | wood | 1 | 1 | 1 | 1 | 1 | 65.3 | 4.0 | 4.0 | 26.7 | NA | NA | NA | NA |
| | | leaf | 1 | 6 | 6 | 6 | 6 | 36.4 | 25.1 | 10.9 | 27.6 | 6.8 | 13.1 | 1.2 | 6.3 |
| | | root | 1 | 2 | 2 | 2 | 2 | 41.7 | 16.9 | 8.4 | 33.0 | 2.4 | 5.5 | 0.3 | 3.3 |
| | *Pinus nigra var. corsicana* (J.W. Loudon) Hyl. | wood | 4 | 22 | 22 | 22 | 22 | 66.6 | 3.3 | 4.0 | 26.1 | 2.9 | 1.5 | 2.4 | 1.3 |
| | | leaf | 2 | 1 | 27 | 27 | 1 | 47.1 | 15.2 | 13.8 | 23.9 | NA | 6.3 | 6.3 | NA |
| | | root | 4 | 10 | 10 | 10 | 10 | 36.0 | 9.2 | 11.9 | 42.9 | 4.9 | 4.4 | 3.1 | 7.3 |
| | *Pinus pinaster* Aiton | wood | 4 | 22 | 22 | 22 | 22 | 66.6 | 3.3 | 4.0 | 26.1 | 2.9 | 1.5 | 2.4 | 1.3 |
| | | leaf | 2 | 1 | 27 | 27 | 1 | 43.2 | 18.2 | 16.5 | 22.1 | NA | 7.5 | 7.5 | NA |
| | | root | 4 | 10 | 10 | 10 | 10 | 36.0 | 9.2 | 11.9 | 42.9 | 4.9 | 4.4 | 3.1 | 7.3 |
| | *Pinus sylvestris* L. | wood | 1 | 1 | 1 | 1 | 1 | 71.7 | 0.9 | 1.0 | 26.4 | NA | NA | NA | NA |
| | | leaf | 1 | 3 | 3 | 3 | 3 | 40.7 | 17.0 | 16.0 | 26.3 | 3.8 | 7.5 | 6.5 | 2.4 |
| | | root | 2 | 4 | 10 | 10 | 4 | 51.2 | 4.4 | 6.0 | 38.4 | 3.7 | 1.4 | 1.4 | 4.5 |

Table 1 Litter carbon quality of the species present in the French RENCOFOR network, estimated based on the literature. In the column "Case," each number corresponds to one case of data availability in the literature: 1 - at least one dataset of complete chemical composition (i.e. for AWEN) exists at species level; 2 - at least one dataset of incomplete chemical composition (only for A, N and the sum of W and E) exists at species level; in this case, the mean proportion of W and E at genus level is used; 3 – no data are available at species level, but at least one complete dataset of chemical composition exists at genus level; 4 - no data are available at species level, but at least one incomplete dataset of chemical composition exists at genus level; in this case, the mean proportion of W and E at tree functional type level is used; 5 – no data are available at neither species nor genus level, in this case, the mean AWEN composition at tree functional type level is used. Cases 1 to 5 in descending order of priority.

| Data | Observed litter input quantity (mean ± SD, in tC ha⁻¹ yr⁻¹) | | Year | | | | | | | | | | | | | | | | |
|---|---|---|---|---|---|---|---|---|---|---|---|---|---|---|---|---|---|---|---|
| | Conifers (51 sites) | Broadleaves (50 sites) | 1961-1990 | 1991 | 1992 | 1993 | 1994 | 1995 | 1996 | 1997-2005 | 2006 | 2007 | 2008 | 2009 | 2010 | 2011 | 2012 | 2013 | 2014 |
| Climate | - | - | M | M | M | M | M | M | M | M | M | M | M | M | M | M | M | M | M |
| Organic matter inputs via forests | | | | | | | | | | | | | | | | | | | |
| Fruits and miscellaneous | 0.36 ± 0.28 | 0.64 ± 0.41 | | | | | M | M | M | M | M | M | M | | | | | | |
| Leaves | 1.12 ± 0.35 | 1.28 ± 0.31 | | | | | M | M | M | M | M | M | M | | | | | | |
| Fine branches | 0.29 ± 0.14 | 0.45 ± 0.14 | | | | | M | M | M | M | M | M | M | | | | | | |
| Coarse woody branches* | 0.32 ± 0.14 | 0.72 ± 0.29 | | | | | M | M | M | M | M | M | M | M | M | M | M | M | M |
| Stems* | 0 | 0 | | | | | 0 | 0 | 0 | 0 | 0 | 0 | 0 | 0 | 0 | 0 | 0 | 0 | 0 |
| Coarse woody roots* | 0.83 ± 0.36 | 1.03 ± 0.38 | | | | | E | E | E | E | E | E | E | M | M | M | M | M | M |
| Fine roots | - | - | | | | | E | E | E | E | E | E | E | | | | | | |
| Soil carbon stock | - | - | | | M | | | | | M | | | | | | | | M | |

Table 2: A summary of the data used for Yasso07 simulations in the present study. In the "Year" columns: M - measured data; E - estimated data ~~according to the~~based on measured ~~ones~~data; 0 – noted, but the contribution to litter is negligible. For soil carbon stock measurements, zones in dashed line ~~zones~~ denote the inventory duration. For each year, each symbol (M and E) only account for the general case, ~~and~~ hence it is possible that measurements ~~was~~ were occasionally omitted at some sites. * - litter input caused by harvest~~g~~ or storms were included (~~once~~ after they had occurred); SD - standard deviation; litter input~~s~~ ~~are~~ is dry matter~~s~~. Diameters used for defining each litter type: ≤2 cm for fine branches, >4 cm for coarse woody branches, > 5 mm for coarse woody roots and ≤ 5 mm for fine roots.

**Figures**

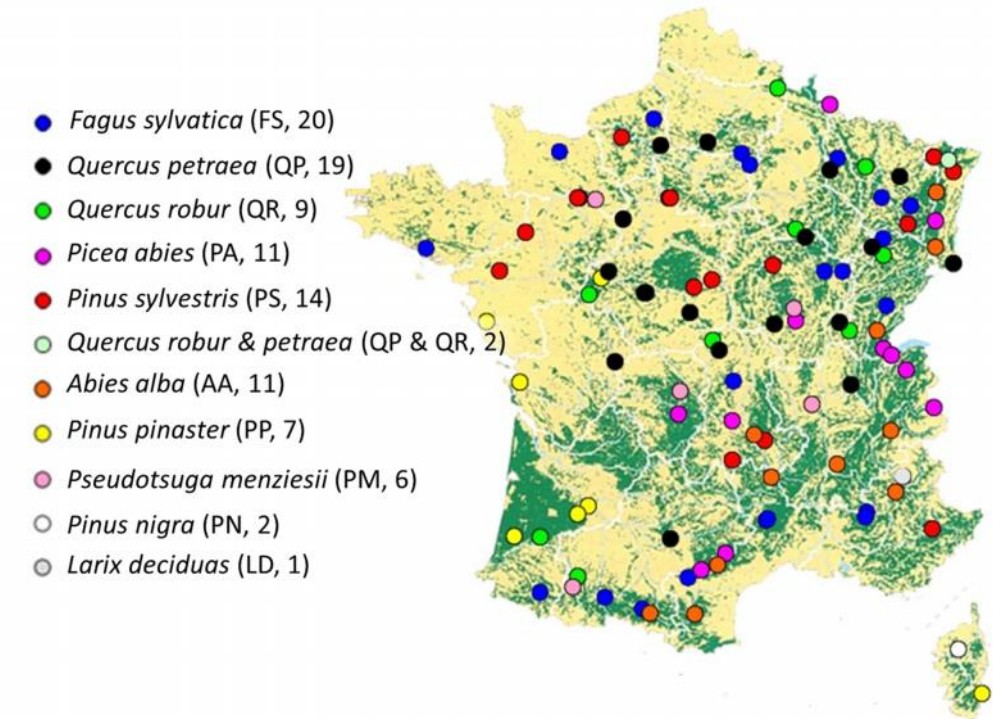

- ● *Fagus sylvatica* (FS, 20)
- ● *Quercus petraea* (QP, 19)
- ● *Quercus robur* (QR, 9)
- ● *Picea abies* (PA, 11)
- ● *Pinus sylvestris* (PS, 14)
- ○ *Quercus robur & petraea* (QP & QR, 2)
- ● *Abies alba* (AA, 11)
- ○ *Pinus pinaster* (PP, 7)
- ○ *Pseudotsuga menziesii* (PM, 6)
- ○ *Pinus nigra* (PN, 2)
- ○ *Larix deciduas* (LD, 1)

Figure 1: Geographical distribution of the sites ~~of~~ in the RENECOFOR network used ~~for~~ to
test~~ing the~~ Yasso07 performance ~~of Yasso07~~ (see also Jonard et al., 2017). Forested areas are
represented in green. Each circle represents one site; the color represents the dominant tree
species ~~of~~ on the plot. In ~~each pair of~~ parentheses, the species abbreviation and number of
sites ~~by~~ per species are indicated.

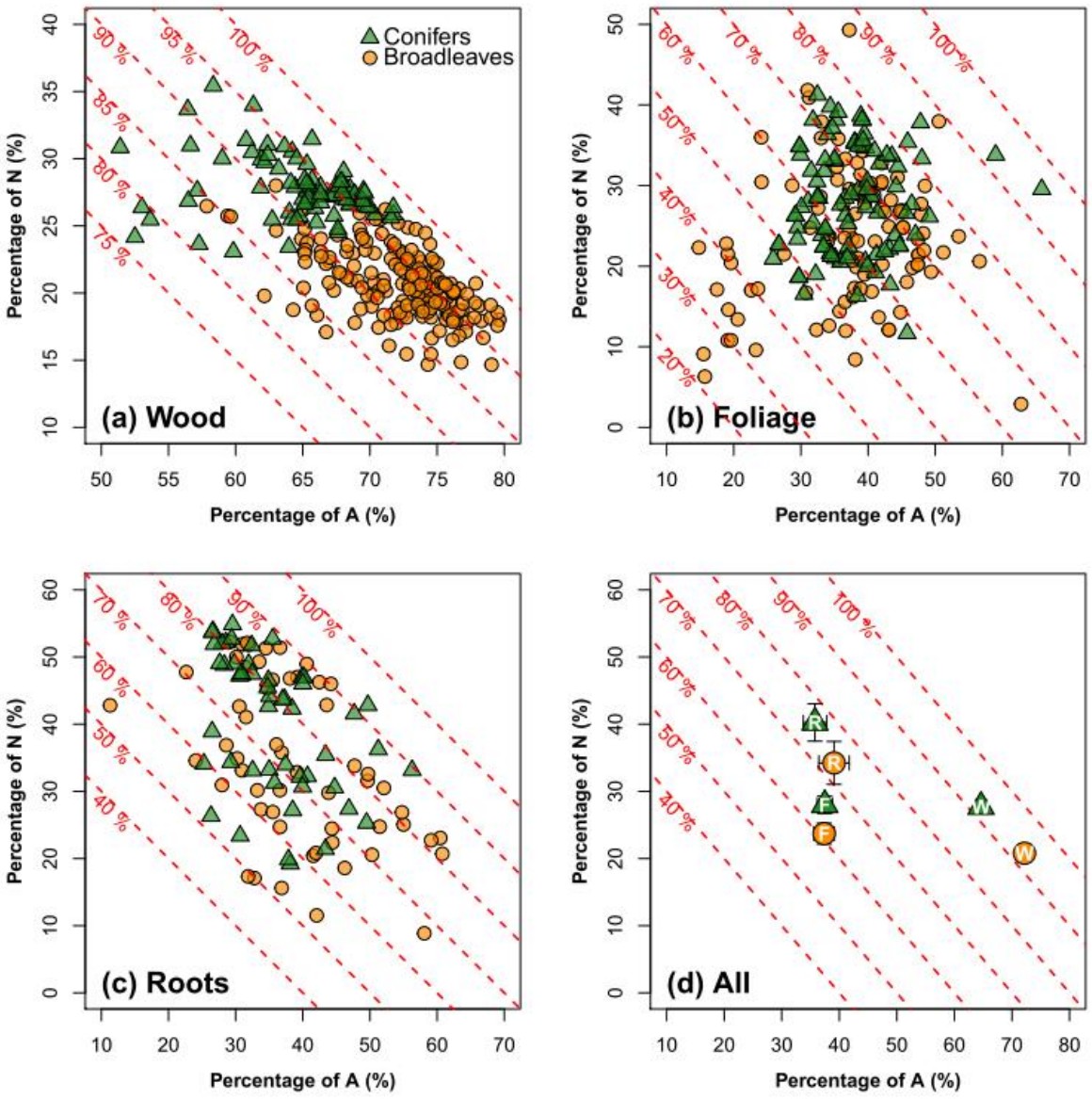

Figure 2: A meta-analysis of the carbon composition for northern temperate tree species: the *x*-axis represents the percentage of acid-hydrolysable compounds (e.g. cellulose, noted ~~by~~ as A, in %) and the *y*-axis represents the percentage of non-soluble and non-hydrolyzable comp~~r~~ounds (e.g. lignin, noted ~~by~~ as N, in %). The oblique dashed red lines ~~notify~~ show the sum of A and N, and their ~~the~~ values ~~of which are shown here~~. The remaining percentage, i.e. 100 - A - N, refers to the portion of ~~compounds like~~ non-polar extractives, ethanol or dichloromethane (E), or compounds in water (W). ~~(a)~~ Analys~~e~~is conducted (a) for wood (106 data points for broadleaves; 79 for conifers), (b) for foliage litter (~~b,~~ 106 data points for broadleaves; 83 for conifers), and (c) for root litter (58 data points for broadleaves; 49 for conifers)~~.~~; (d) is a statistical ~~synthesis~~ summary (symbols – means and error bars – 1.96 * standard error) ~~of~~ for wood (W), foliage (F) and roots (R) in a common coordinates system. ~~Attention to~~Note the use of different axis graduations in each plot. See Supplementary Material II for the data sources. Note the different y-axis scales.

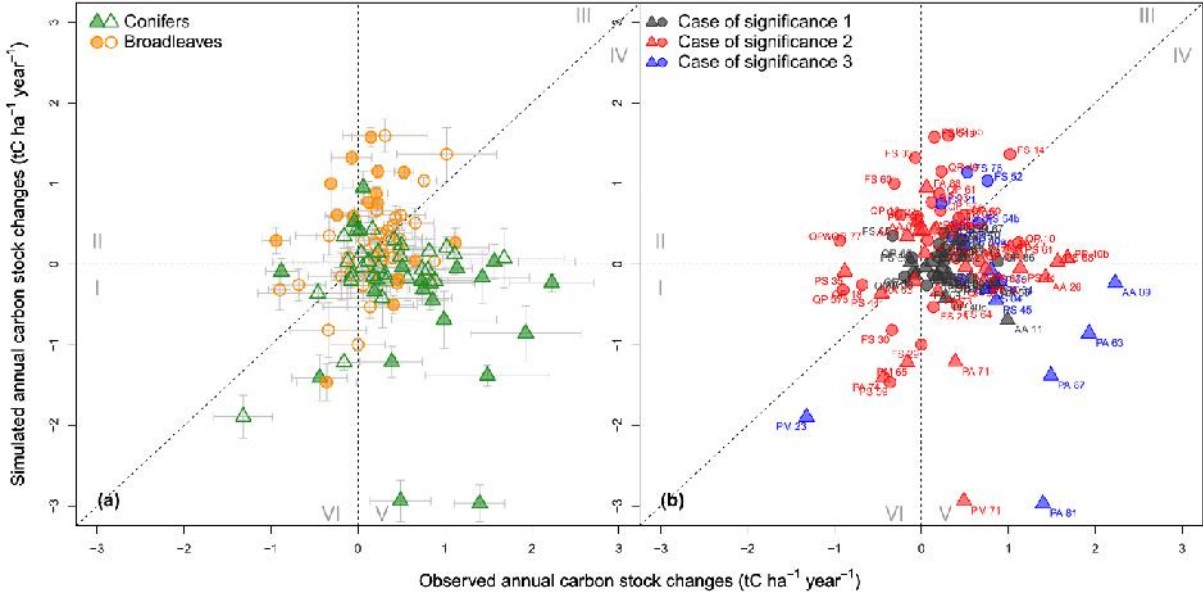

Figure 3: Comparison between simulated and observed ~~changes in~~ annual carbon stocks ~~changes~~ (ACC, in tC ha$^{-1}$ year$^{-1}$). ~~Round~~ Circles and triangles respectively ~~symbols~~ represent sites dominated by broadleaves and conifers~~, respectively~~. ~~P~~The partial steady-state assumption was used ~~for~~ when initializing carbon stock quality ~~of the stock until~~to 1.0 m in depth. ~~The chosen fine~~ The fine-root:foliage ratio for broadleaves and conifers is 1.0. To facilitate ~~discussions~~readability, ~~we set~~ Roman ~~numbers~~ numerals (I-VI) denot~~eing~~ the six zones in which data points are distributed. In (a), error bars represent standard errors; hollow and filled points respectively represent non-significant and significant differences between simulated and observed ACC according to t-test (at a 95% confidence ~~level~~interval). In (b), case of significance: 1 – no significant difference from 0 for ~~n~~either observed ~~n~~or simulated ACC; 2 - a significant difference from 0 for either observed or simulated ACC and 3: - a significant difference from 0 for both observed and simulated ACC.

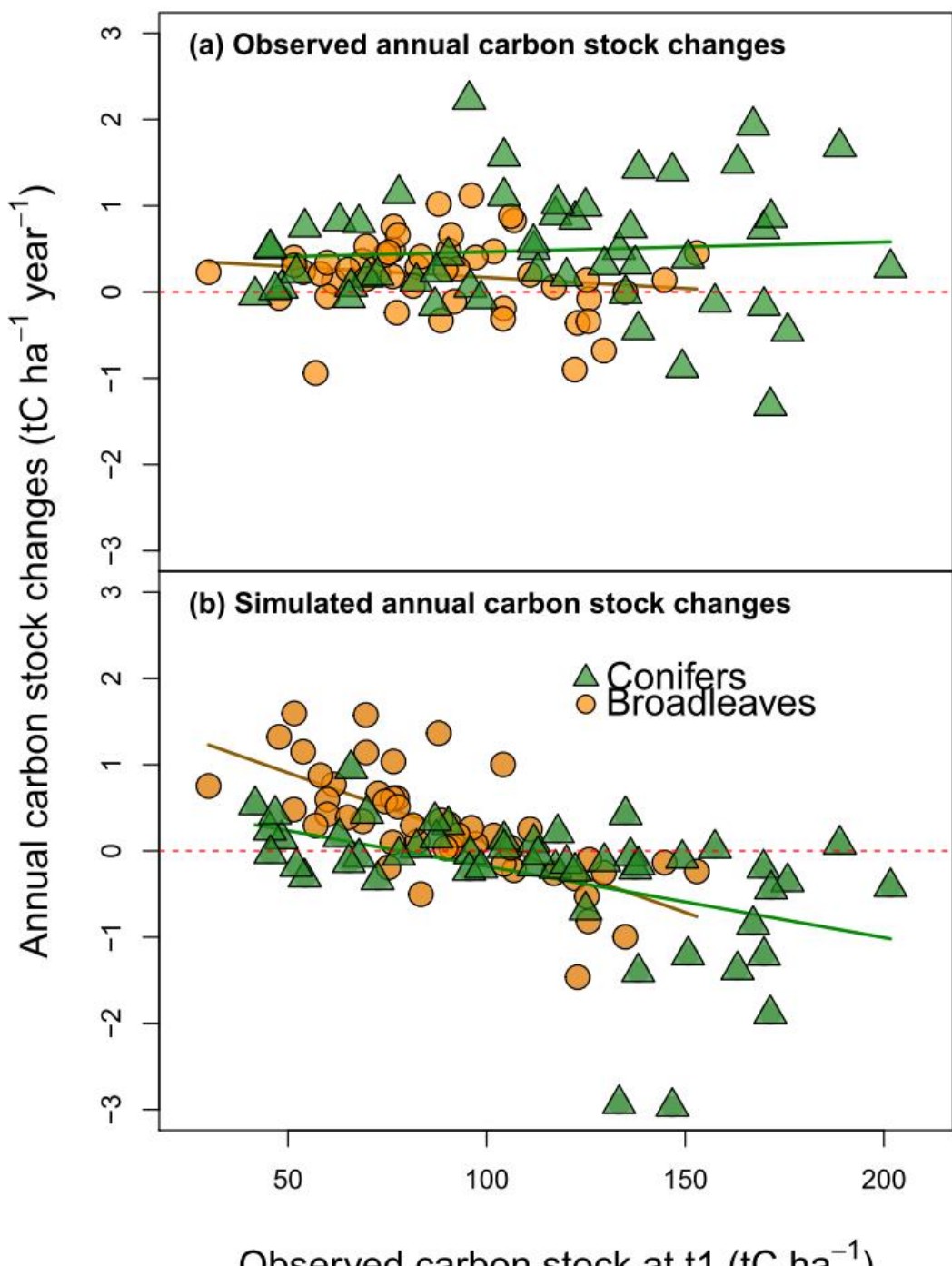

Figure 4: Observed (*y*-axis, a) and simulated annual ~~change~~ changes in carbon stocks (*y*-axis,
b) plotted against the ~~observed~~ carbon stocks observed ~~until~~ to 1.0 m in depth (*x*-axis) during
the first soil carbon stock ~~inventory~~assessment. Regressions: $y = -0.003x + 0.422$ ($R^2 = 0.03$)
for observed values ~~in~~ at the sites dominated by broadleaves; $y = 0.001x + 0.353$ ($R^2 = 0.01$)
for observed values at the sites dominated by conifers; $y = -0.016x + 1.715$ ($R^2 = 0.62$) for
simulated values ~~of~~ for the sites dominated by broadleaves; $y = -0.008x + 0.648$($R^2 = 0.60$) for
simulated values ~~of~~ for the sites dominated conifers.

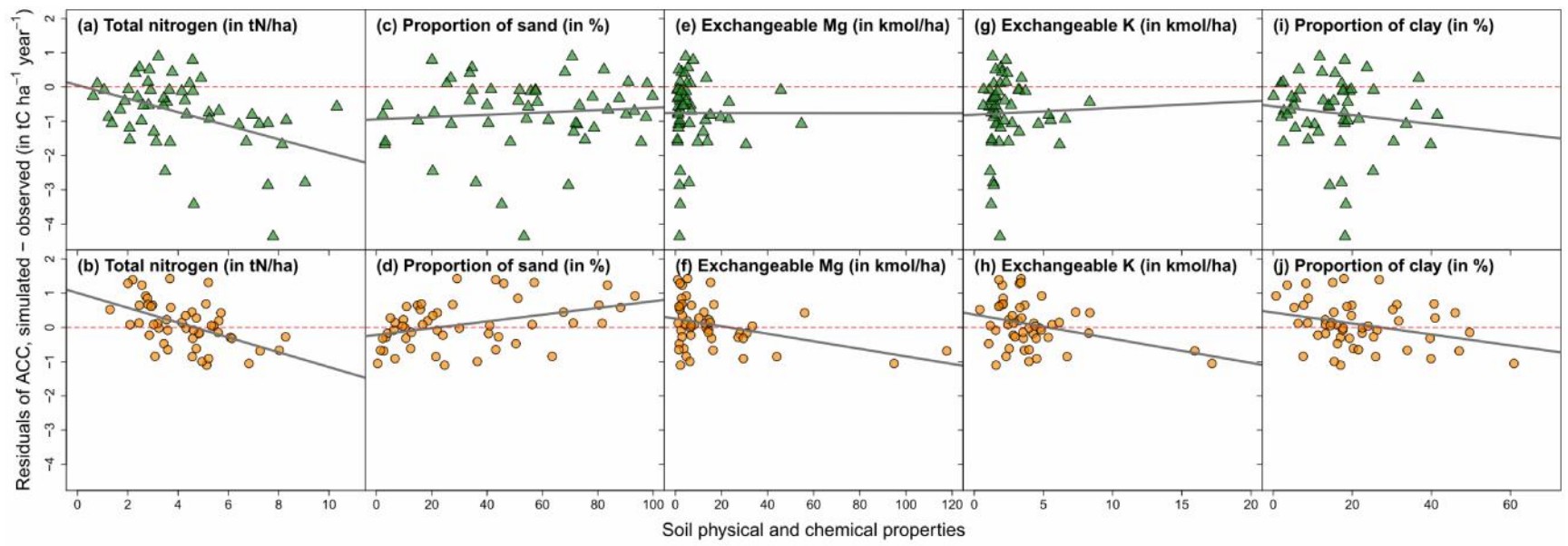

Figure 5 Residuals plotted against selected soil physical and chemical properties. Top plots with green triangles ~~stand for~~represent ~~the~~ sites
dominated by conifers and bottom plots with orange dots ~~stand for~~represent ~~the~~ sites dominated by broadleaves. Regressions in all ~~the~~ five
subplots for the broadleaved sites (b, d, f, h and i) and in one subplot for the stands dominated by conifers (a) are significant (P<0.5*). See Table
S2 for linear regression results ~~of linear regressions of~~ for all ~~the~~ 11 soil variables. Red dashed line indicates the zero line.

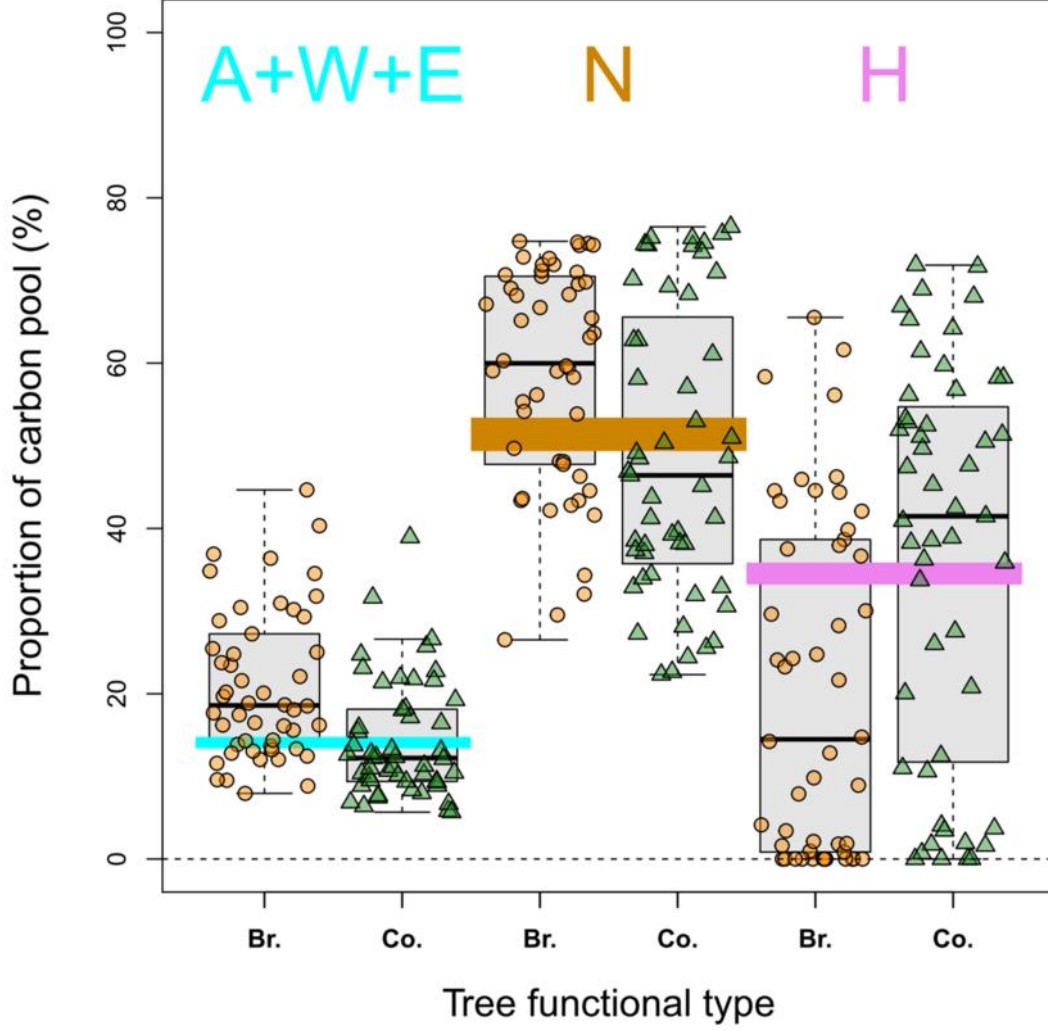

Figure 6: Distribution of estimated carbon qualityies based on the partial steady-state
assumption (boxplots) versus those based on the complete steady-state assumption (whose
ranges are all very narrow and are expressed with strips in colourcolor: 13 – 15 % for the sum
of A, W and E (cyan); 49 – 53 % for N (brown); 33 – 36 % for H (purple)). For each boxplot,
the lower and top edge of the box corresponds to the 25th and 75th percentile data points,
respectively; the line within inside the box represents the median; there are no outlier points in
this case. Br. – Broadleaveds stands; Co. – Coniferous stands.

**Supplementary Materials**

**Supplementary Materials I:** Supplementary tables and figures.

**Supplementary Materials II:** Database for the meta-analysis of wood and litter chemical composition.