# Peer review of "Modeling soil organic carbon dynamics in temperate forests ~~using~~ with Yasso07"

_Biogeosciences, 2018_

## Referee Comment (RC1) · Anonymous Referee #2 · 6 Jul 2018

Mao et al. submitted an interesting manuscript about the evaluation of the Yasso07 model against RENECOFOR dataset a French network of forest plot. The paper is generally well written and the methodology sounds. It also fits well with the Biogeosciences scope. Nevertheless, the main message of the manuscript which seems to be that Yasso07 may not be the best tool to evaluate soil carbon changes in forest when it used outside the context of boreal forest where it has been originally developed is a bit diluted because the manuscript is too long. In particular, I suggest moving the sensitivity analysis in supplementary material (Fig 6 to 8). Regarding the sensitivity analysis I did not fully understood the Module II and the interest to test effect of simulation length; this should be removed or better explained.

Minor comments: P2 L3 I am not sure Yasso represents the whole state of the art.

Some mechanisms are missing and it has a humus pool whereas the humus concept is now criticized (Lehmann, J. & Kleber, M. 2015)

P6 L3-4: Are that information not available in the ICP forest network?

P8 L12-13: In the original dataset to calibrate the model is there some data coming from RENECOFOR sites?

P13 in eq. 7 the second line of the equation should be ACCsim=(CSsim,t2 – Cssim,t1)/(t2-t1), right? If not please better explained, if yes please check that this only a typo mistake and the calculation were made the good way.

Table 2: is 'ignorable' the good terms do you mean negligible?

Fig. 3: Please don't call the non-hydrolysable compounds N. It is a misleading acronym since it is more used for nitrogen.

---

## Referee Comment (RC2) · T. Wutzler (Referee) · 30 Jul 2018

The study presents a model- data comparison at multi-site scale of fores sites which are relevant for management policies and accounting for global climate negotiations.

The presentation is good and I could understand what has been done. Especially the litter quality database part is already valuable to other scientists. For the model-data comparison I have several remarks of what should be done additionally/differently, that potentially could alter the conclusion quite severely. Because of the paper did not change much compared to the pre-public-discucssion, I repeat my comments in the this public discussion.

1) Steady state and observed stocks: The authors computed litter quality (percent-

ages) from steady state computations and then scaled all pools down so that the sum matched observed initial stocks. Assuming that lower stocks resulted by recovery from disturbance, however, the composition of the faster pools should be closer to steady state than the slow pools. I recommend repeating the simulations with an additional initialization procedure according to Wutzler 2007.

2) Comparing different soil depths: The authors argue that stock changes are less susceptible to differences in soil depth than stocks, because the more stable pools reside in deeper layers. However, they did not account for this effect on initialization of stock qualities. I suggest instead transforming the observations (down to 1m) to the depth assumed by the YASSO model (0.4m) before comparison. This should be possible, because several depths were measured, e.g. by fitting a function to the depth distribution of bulk density and carbon concentrations and computing the cumualive stock up to a certain depth.

3) Effects of mineralogy and potential stocks: The authors did not explain variation in residuals well by studied explanatory variables. I suggest including some soil mineralogy measures. Additionally, one could include potential stocks as derived from mineralogy by Feng 2013 and Beare 2014 or the indicators by Rasmussen 2018 to include a measure of distance to potential.

General comments (locations refer to the pre-public-discussion version)

p3l25: The authors claim that at annual time aggregation, first order decomposition is adequate. However, largest criticism of first order comes from interaction among pools, like priming instead of time aggregation (Wutzler 2013)

p4l5: The authors claim to be first study of larger scale YASSO application. I know that YASSO is the soil model of the MPI earth system model implemented by Tea Thum, and suspect that there should be also larger scale studies.

Sect. 3.4 and complicated figure 8 express the simple fact that there are initially high

changes and later on slower changes in recovering C-Stocks. They can be shortened very much.

References

Beare M, McNeill S, Curtin D, Parfitt R, Jones H, Dodd M & Sharp J (2014) Estimating the organic carbon stabilisation capacity and saturation deficit of soils: a New Zealand case study. Biogeochemistry, Springer Science + Business Media, 10.1007/s10533-014-9982-1

Feng W, Plante A & Six J (2013) Improving estimates of maximal organic carbon stabilization by fine soil particles. Biogeochemistry, Springer Science + Business Media, 112, 81-93 10.1007/s10533-011-9679-7

Rasmussen C, Heckman K, Wieder W, Keiluweit M, Lawrence C, Berhe A, Blankinship J, Crow S, Druhan J, Pries C, Marin-Spiotta E, Plante A, Schädel C, Schimel J, Sierra C, Thompson A & Wagai R (2018) Beyond clay: towards an improved set of variables for predicting soil organic matter content. Biogeochemistry, Springer Nature, 137, 297-306 10.1007/s10533-018-0424-3

Wutzler T & Reichstein M (2007) Soils apart from equilibrium – consequences for soil carbon balance modelling. Biogeosciences, 4, 125-136 10.5194/bg-4-125-2007

Wutzler T & Reichstein M (2013) Priming and substrate quality interactions in soil organic matter models. Biogeosciences, 10, 2089-2103 10.5194/bg-10-2089-2013

---

## Author Comment (AC1) · 31 Aug 2018

Dear Referee,

Please find the attached zip file containing our responses to your review.

- responses to comments: "20180831_reponses.pdf" - revised manuscript with traces: "20180831_MaoETAL_Yasso07_withTraces.pdf" - revised manuscript without traces: "20180831_MaoETAL_Yasso07_withoutTraces.pdf" - revised supplementary material I: "20180831_MaoETAL_Yasso07_SupplementaryMaterial-I.pdf"

Thank you very much again for your expertise ! Best wishes,

Mao et al.

Please also note the supplement to this comment:
https://www.biogeosciences-discuss.net/bg-2018-219/bg-2018-219-AC1-
supplement.zip

---

## Author Comment (AC2) · 31 Aug 2018

Dear Thomas,

Please find the attached zip file containing our responses to your review.

- responses to comments: "20180831_reponses.pdf" - revised manuscript with traces: "20180831_MaoETAL_Yasso07_withTraces.pdf" - revised manuscript without traces: "20180831_MaoETAL_Yasso07_withoutTraces.pdf" - revised supplementary material I: "20180831_MaoETAL_Yasso07_SupplementaryMaterial-I.pdf"

Thank you very much again for your expertise !

Best wishes,

[Figure]

Mao et al.

Please also note the supplement to this comment:
https://www.biogeosciences-discuss.net/bg-2018-219/bg-2018-219-AC2-supplement.zip

---

## Author Response (AR1)

**General words to Referees.**

We truly thank both referees for their generally positive comments and highly valuable suggestions toward the manuscript. Here, we greatly appreciate their expertise.

We are providing our responses to these comments point by point as shown below. We have to admit that the corrections we are making are a product of compromise between referee suggestions. Referee 2 points out that the manuscript is too long and suggests significantly shortening and simplifying the manuscript (for example, removing the sensitivity analysis II), while Referee 1 proposes adding extra simulations and new data for testing new ideas. Given such a fact, we tried to make a rational decision by considering multiple factors while keeping the manuscript well-balanced.

We hope the revised version is a good one that can both meet Referees' expectations and the journal's criteria.

PS: The number of page and lines mentioned below correspond to those in the **version with traces**.

- **Referee 2, anonymous**

**Referee 2**: Mao et al. submitted an interesting manuscript about the evaluation of the Yasso07 model against RENECOFOR dataset a French network of forest plot. The paper is generally well written and the methodology sounds. It also fits well with the Biogeosciences scope.

**Authors**: Thank you for this positive remarks !

**Referee 2**: Nevertheless, the main message of the manuscript which seems to be that Yasso07 may not be the best tool to evaluate soil carbon changes in forest when it used outside the context of boreal forest where it has been originally developed is a bit diluted because the **manuscript is too long**. In particular, I suggest **moving the sensitivity analysis in supplementary material (Fig 6 to 8)**. Regarding the **sensitivity analysis, I did not fully understood the Module II and the interest to test effect of simulation length**; this should be removed or better explained.

**Authors**: We are aware of the current length of the manuscript and also the visually complex Figures associated with the Sensitivity Analyses (SA), especially the Module II (although their originality sounds). To better focus on RENECOFOR data fit to Yasso07, now we decide to:

(i)     move the initial Fig. 6 (one of the two figures corresponding to the SA – Module I) and initial Fig. 8 (the only figure corresponding to the SA – Module II) to Supplementary Materials (SM), see the **Fig. S8** and **Fig. S9** in SM1.

(ii)    replace initial Fig. 7 (boxplots corresponding to SA Module I) by a more understandable **Fig. 6**, which shows steady-state carbon quality as a function of initial carbon stock for all the 101 RENECOFOR sites.

(iii)   simplify, accordingly, the descriptions to these results, see **Sect. 3.3, P18** and **Sect. 3.4, P 18**.

(iv)    simplify and clarify the description of SA in Materials and Methods, see **Sect. 2.6.1** and **Sect. 2.6.2, P13**.

Besides, we also made an effort to shorten the manuscript, when it is necessary and possible. For example, we also put the initial Fig. 1 (Yasso07's model structure) in Supple. Mat, see **Fig. S1**. This is because Yasso and Yasso07 are fairly well-known and many papers working on Yasso07 do not necessarily show such a figure. Reducing those above text and figures also provides us opportunities for adding two analyses suggested by Referee 1 without too expanding the length of the manuscript.

**Referee 2**: Minor comments: P2 L3 I am not sure Yasso represents the whole state of the art.

**Authors**: Sorry for this ambiguity of expression. We would mean the issues encountered in the use of Yasso07 are representative ones in the current modelling of soil carbon dynamics. So now we rephrase this sentence using "current bottleneck" instead of "the state of the art", see **P2, LN1-6**: "*We revealed, taking YASSO07 as model support, the current bottleneck of soil carbon modelling due to lacking knowledge or data on soil and litter carbon quality and fine root litter quantity, rendering high uncertainties for model inputs.*"

**Referee 2**: Some mechanisms are missing and it has a humus pool whereas the humus concept is now criticized (Lehmann, J. & Kleber, M. 2015)

**Authors**: Yes, but pool based models are still prevailing ones widely used in research and development. We decide to add this information and also the reference to remind readers when introducing the "H" pool, see **P7, LN9-11**.

**Referee 2**: P6 L3-4: Are that information not available in the ICP forest network?

**Authors**: We should say it is hard and extremely rare to obtain such a national scale dataset that contains such complete information (climate, soil with time series, fine and coarse litterfalls with time series) which are usually done on very instrumented sites (not national networks). Many countries involved in ICP Forests have such data but the main strength of the RENECOFOR network is to have, for two soil surveys, data obtained with exactly the same methods making estimation of SOC change possible. As far as we know, similar data also exist in some countries out of Europe (e.g. in China), but still remain inaccessible to us.

To avoid being too absolute, we decide to delete the sentence, see **P6, LN5-6**, since we have already highlighted the rarity of the dataset before, see **P5, LN30**.

**Referee 2**: P8 L12-13: In the original dataset to calibrate the model is there some data coming from RENECOFOR sites?

**Authors**: no, because Yasso07 was first published in the year of 2009 (Tuomi et al., 2009), i.e., the year when the RENECOFOR's 2nd soil inventory campaign was still ongoing. The dataset was first published in 2017 (see Jonard et al. 2017) and this is the first time that the dataset is used for testing Yasso07.

**Referee 2**: P13 in eq. 7 the second line of the equation should be ACCsim=(CSsim,t2 –Cssim,t1)/(t2-t1), right? If not please better explained, if yes please check that this only a typo mistake and the calculation were made the good way.

**Authors**: After checking, our equation should be the right one, because we used the observed C stock at t1 (CSobs, t1) as the input to simulate the C stock at t2 (CSsim,t2).

Now, following the suggestion given by Referee 1, we also performed simulations to calculate the stock until 1 meter and had this steady-state stock value ($CS_{steady-state}$) compared with $CS_{obs,t1}$ down to a depth of 1 meter. Please see below and also the text. See also **Fig. S4**.

**Referee 2**: Table 2: is 'ignorable' the good terms do you mean negligible?

**Authors**: Done. "negligible" is now used, see **P35, LN10**.

**Referee 2**: Fig. 3: Please don't call the non-hydrolysable compounds N. It is a misleading acronym since it is more used for nitrogen.

**Authors**: we did notice this potentially misleading term, but we think that it is more important to follow the Yasso07 inventors' given terminology. This allows keeping consistency among studies working on Yasso07 and facilitating inter-study comparisons. Moreover, in the case of this paper, we don't think the use of "N" can be really misleading, as "nitrogen" was always fully spelled when appearing in the main text.

We decide to add a note in the table of "Nomenclature and abbreviations", saying that in none of case "N" means nitrogen in this paper and when nitrogen is mentioned (for example in Figure. 5 and Figure S7), we used "nitrogen" , not "N". See **Page 3**. In order to avoid too many acronyms, we checked the text and kept using "carbon" instead of "C."

- **Referee 1 (R1), T. Wutzler (twutz@bgc-jena.mpg.de)**

**Referee 1 :** The study presents a model- data comparison at multi-site scale of forest sites which are relevant for management policies and accounting for global climate negotiations. The presentation is good and I could understand what has been done. Especially the litter quality database part is already valuable to other scientists.

**Authors**: Thank you Thomas for your positive remarks!

For the model-data comparison I have several remarks of what should be done additionally/differently, that potentially could alter the conclusion quite severely. Because of the paper did not change much compared to the pre-public-discucssion, I repeat my comments in the this public discussion.

Authors: Sorry for the delayed responses, as it took us some time to obtain the new soil data from the network and to perform additional analyses.

**Referee 1:** 1) Steady state and observed stocks: The authors computed litter quality (percentages) from steady state computations and then scaled all pools down so that the sum matched observed initial stocks. Assuming that lower stocks resulted by recovery from disturbance, however, the composition of the faster pools should be closer to steady state than the slow pools. I recommend repeating the simulations with an additional initialization procedure according to Wutzler 2007.

**Authors**: The alternative method, i.e. the relaxed equilibrium assumption (REA) method, proposed in Wutzler (2007) is indeed very interesting and should definitely be better highlighted in our manuscript (see below). However, we do have concerns of applying such a method to this manuscript. We don't think that, until now, we've really have enough information to repeat the simulations using such an approach. How can we properly choose the current rate of assimilation (delta_Cc/delta_t in Eq. 4) that might be site- or specific dependent? Shouldn't we still need to make some critical assumptions? With a changed AWENH composition, the results would probably be different (as showed in our sensitivity analysis Fig. 8, now Fig. S9 in Supple. Mat.), but would they be more reliable?

Even though we can do extra- sensitivity analyses to justify all the above things, but wouldn't all this make the manuscript's objective too diffuse, even shifted? For us, testing the regular and REA methods (just like the work performed in Wutzler (2007)) can be totally an independent study which corresponds to a new paper. When saying this, here we should add that, actually, we are indeed conceiving a new paper project tackling the issue of soil carbon quality initiation. Specifically, we aim to re-simulate the RENECOFOR sites' C dynamics in Yasso07, by using the site- and depth- dependant composition of carbon in different ages (determined by the 14C method, analysis still ongoing), instead of the regular initialisation methods. This project follows the idea of the newly published paper (Balesdent et al. (2018) Nature 559, p.599–602) that showed vertical heterogeneity of composition of carbon age along soil profiles. Also, this project's idea is in line with the hypothesis of the REA method, i.e., soil carbon quality may not be set as that at its perfectly steady state in theory.

Despite such a choice of not doing REA simulations and associated sensitivity analyses, we've decided to **expand our discussion** regarding this point. First, we cited this work and highlighted the existence of this method that merits more attention. Thus, we proposed therefore to perform an independent study on the test of different initialisation methods by using different pool-based carbon models (Yasso, Yasso07, RothC etc.), as no such work has been done so far.

Additionally, we further pointed out that solely testing different methods of model initialisation, does not allow radically solving the uncertainty issue. We propose therefore considering specific or generic curves of carbon age ~ soil depth + ecosystem type in the future carbon dynamics modelling, following the key message of Balesdent et al. (2018).

Please see these added discussions in **Sect. 4.2, P22, LN 15 –26** and **P23, LN 7 –12.**

We hope you can understand such a decision we made with compromises and appreciate the improvements in the current version.

**Referee 1** : Comparing different soil depths: The authors argue that stock changes are less susceptible to differences in soil depth than stocks, because the more stable pools reside in deeper layers. However, they did not account for this effect on initialization of stock qualities. I suggest instead transforming the observations (down to 1m) to the depth assumed by the YASSO model (0.4m) before comparison. This should be possible, because several depths were measured, e.g. by fitting a function to the depth distribution of bulk density and carbon concentrations and computing the cumualive stock up to a certain depth.

**Authors:** Thank you for this suggestion.

We contacted RENECOFOR and, fortunately, obtained the ground truth data of soil density for the depth of 40-100 cm for each site, although these data do not have the 5-subplot replicates as the 0-40 cm ones. Now we have estimated the carbon stock until a depth of 1 m based on some these additional data.

Following your suggestion, we now are able to compare the observed C and simulated carbon stock until a depth of 1 meter. Because of the length of the manuscript (which is the major criticism of Referee 2) and absence of replicates for 40-100 cm, we still would like to focus mainly on carbon change (ACC) as our major objective rather than on carbon stock (CS). But the latter can indeed be considered a good way of checking Yasso07's theoretical prediction. Running this simulation also gives us a good opportunity to show RENECOFOR site-dependent steady-state carbon quality, which is shown in a new **Fig. 6** replacing the old boxplot Fig. 7.

Accordingly, we put the plot related to carbon stock in Supple. Materials (see **Fig. S4**) and gave descriptions in Results (see **P16, LN26-30** and **P18, LN16-19**). Certainly, we also added related information in M&M on observational data of 40-100 cm (see **Sect. 2.2.1, P9, LN2-17**) and simulation (see **P14, LN22-27**).

**Referee 1**: Effects of mineralogy and potential stocks: The authors did not explain variation in residuals well by studied explanatory variables. I suggest including some soil mineralogy measures. Additionally, one could include potential stocks as derived from mineralogy by Feng 2013 and Beare 2014 or the indicators by Rasmussen 2018 to include a measure of distance to potential.

**Authors:** Indeed, soil texture and mineralogy greatly affect soil biogeochemical cycling and carbon stock. Follow this idea and your valuable suggestion, we contacted RENECOFOR and obtained a new dataset including soil physical (texture) and chemical (pH, stocks of total nitrogen, total phosphorus, exchangeable Al, K, Ca and Mg) of the 101 sites.

We added these variables to the residual analyses. We added a new table in Supple. Mat. For the linear regression results for all of the 11 variables (See **Table S2**).The associated PCA in Supple. Mat. has been updated (see **Fig. S7**). Further, in the main text, we added a new plot about effect of soil properties on residuals as **Fig. 5**. Associated result descriptions and discussions concerning these added results can be found in the main text, see **P16, LN32-P17,LN5**, **P20,LN1 -21** and **P27, LN20-22** and **P27, LN29-31**.

General comments (locations refer to the pre-public-discussion version)

**Referee 1**: p3l25: The authors claim that at annual time aggregation, first order decomposition is adequate. However, largest criticism of first order comes from interaction among pools, like priming instead of time aggregation (Wutzler 2013)

**Authors:** Adding pool interactions will alter Yasso07's fundamental configuration and this is no more the major purpose of the manuscript. So we highlighted this point in the text by citing this work to draw future readers' attention, see **P4, LN21-23**.

**Referee 1**: p4l5: The authors claim to be first study of larger scale YASSO application. I know that YASSO is the soil model of the MPI earth system model implemented by Tea Thum, and suspect that there should be also larger scale studies.

**Authors: in P4LN15, we've used the word "rarely" to avoid to being too absolute.** We also deleted the statement to avoid confusion, see **P6, LN5-6**.

**Referee 1**: Sect. 3.4 and complicated figure 8 express the simple fact that there are initially high changes and later on slower changes in recovering C-Stocks. They can be shortened very much.

**Authors:** We now have decided to move this figure to Supple. Mat, following the suggestion given by Referee 2. Accordingly, the Section 3.4 are shortened, see **Sect. 3.4, P18-19.**

**Referee 1**:

[revised manuscript text omitted]

Regarding soil physical and chemical properties, total nitrogen stock soil were significantly correlated with residuals for both broadleaved and coniferous stands (Fig. 5). Then, soil texture (proportions of clay and sand) and exchangeable magnesium, calcium and potassium were significantly correlated with residuals only for broadleaved stands (Fig. 5; Table S2). The remaining tested variables, such as proportion of silt, pH, total phosphorus and carbon:nitrogen ratio, had no relationship with the residuals, except for exchangeable aluminum, which showed a weak correlation with ACC residuals ($P<0.05$*) only for coniferous stands (Table S2).

**3.3 Effect of litter carbon quality on model prediction (Sensitivity analyses 2.6.1)**

Variation of litter carbon quality (without distinction of original organ) altered the carbon quality at steady-state  (Fig. S8). The  of soil A, W and E carbon pools remained below 15% regardless  the biochemistry of litter inputs. The percentages of  N and H pools were more susceptible to the variation of litter carbon quality than the more labile ones (e.g., A, W and E; )).

The strong sensitivity of the carbon steady state distribution to litter carbon quality was *de facto* greatly discounted in reality, because the variation in chemical composition of tree species was very limited (Fig. 2). This can also be represented by the quite stable and narrow variations of the proportion of soil pools at steady-state for all the 101 RENECOFOR sites (Fig. 6), with the sum of A, W and E pools around 15%, N pool around 55% and H pool around 30-35 %. ~~Using average compositions of broadleaves and conifers species, we found that, at the steady- state, the H pool contains 30 – 40% of soil carbon, the N pool 45 to 55 %, the A pool <5% and W and E pools <2% (Fig. 7). Broadleaves dominated sites differed from conifers dominated sites with a slightly lower percentage N carbon in the steady state soil carbon stock, but a higher percentage of H carbon (Fig. 7).~~

**3.4 Impact of initial condition of soil carbon stock on model prediction (Sensitivity analyses 2.6.2)**

Fig. S98  visualized all the theoretically possible final carbon stocks by varying initial carbon stocks and simulation length (from 1 to 10 000 years). The initial soil carbon quality had a pronounced impact on the final soil organic carbon stocks  at annual and decennial scales. For example, when the initial proportion of A pool increased from 0 to 80%, the final proportion of A could increase by +30 to +40 tC ha$^{-1}$ (Fig. S9a) and the final total carbon stock could decrease by c.a. -20 to -30 tC ha$^{-1}$(Fig. S9u) at annual  and decennial  scales. When simulations were performed over millennium timescale, the initial soil carbon quality did not impact the final soil carbon quality anymore. In other words, the same final soil carbon quality was obtained regardless what the initial soil quality was (Fig. S9).

**4 Discussion**

**4.1 Agreement between simulated and observed annual soil carbon stock changes**

Testing widely popularized soil carbon models using large dataset is highly meaningful work that enables not only assessing the model's ability over various climatic and ecosystem types, but also providing lessons and implications for future modelling work. Here, based on the observed carbon stock data to 1 m soil depth from the RENECOFOR network, , we found the simulated and observed carbon stocks ($CS_{steady\text{-}state}$ versus $CS_{obs,\ t1}$) to 1 m showed the same order of magnitude, validating Yasso07's good capability to predict carbon stock in average at the scale of the French territory. Such good performance at the national scale is consistent with Yasso's aim for generality and supported by previous studies (see Ortiz et al. 2013; Lehtonen et al. 2016; Hernández et al. 2017).

Then, based on the observed annual soil carbon stock changes (ACC) with average 15-year interval between the two inventories, we found the simulated ACC using Yasso07 were significantly biased for more than one third of the French RENECOFOR sites. Particularly, Yasso07 generally overestimated the ACC at the broadleaved stands located in the north of France (Fig. S6a-d) and the overestimation can be exacerbated with lower precipitation. Yasso07 tended to underestimate the ACC in our coniferous stands. Nevertheless, we would expect slightly better performance of Yasso07 in coniferous stands than in broadleaved ones, since the model's estimates have shown good correspondence to measurements (of stocks and/or changes) in coniferous forests, especially the Nordic boreal ones (e.g., Karhu et al., 2011; Ortiz et al., 2013). Except for tree functional type and geographical location (e.g. latitude, which is correlated with climatic variables), qualitative ecological variables that are assumed as key factors influencing carbon sequestration processes, e.g. soil type (except for coniferous stands), storm damage and stand age range,  showed limited  tendencies in explaining residuals. Note that those factors were not fully crossed in the 101 sites, rendering testing each signer factor difficult.

The simulated ACC by Yasso07 showed strongly negative correlation with the observed initial soil carbon stock ($CS_{obs,t1}$), with an overestimation of ACC at sites of lower $CS_{obs,t1}$ and an underestimation at sites of higher $CS_{obs,t1}$  (Figs. 4 and S7). Such phenomenon can be logically explained by the model's mechanism: With increasing initial carbon stock, due to the fairly stable steady-state carbon quality (Fig. 6), there is an increase in the quantity of those easily decomposable compounds, i.e. A, W and E, in soil, which triggers a more substantial mass loss at a decennial scale. However, the observed data on carbon stock changes did not support this trend, suggesting that Yasso07's configuration tends to penalize too much the loss of labile carbon at decennial scale. Compared to broadleaved stands, the slightly steeper slope for coniferous stands in Fig. 4b might be attributed to their higher steady-state proportion of the extremely labile pools (A, W and E) in soil at a given soil carbon stock (Fig. 6a) due to the higher proportion of A, W and E pools in the litter quality of broadleaves (Fig.2).

Several soil physical and chemical properties showed clear correlations (especially for broadleaved stands) with ACC residuals (Fig. 5). Also, in the principle component analyses (Fig. S7), the arrows standing for soil variables are generally closer to the pivoting axis of "initial carbon stock – ACC residuals" than those standing for climatic and geographic variables. The correlations (Table S2 and Fig. S7) may indicate that texture and nitrogen content contribute to lower ACC for broadleaved stands compared to model predictions and that aluminum and perhaps also pH (Fig.S7) could be involved in the mechanisms that allow increasing microbial activities and carbon mineralization in soils of coniferous stands compared to model predictions. All these results suggest a potential interest of incorporating soil properties into new versions of Yasso model family, in which soil parameters are lacking or only implicitly incorporated. Indeed, there are numerous evidences that soil physical and chemical properties can greatly govern soil carbon dynamics and stock capacity (Beare et al., 2014; Dignac et al., 2017; Rasmussen et al., 2018),

The limitations of the model at the site-scale are not surprising as the model was developed for primarily large-scale application integrating processes that dominate at the site scale. Despite Yasso07's significant prediction bias at a number of sites, it is unreasonable to simply attribute the bias to the model *per se*, as multiple uncertainties affecting the quality of the model's input data can be identified (see Sects. 4.2 – 4.4). These uncertainties can occur not only with Yasso07, but also with other prevailing models one may choose, highlighting large knowledge gaps in ecology and soil carbon modelling.

**4.2 Soil carbon quality: a recurrent challenge in soil carbon modelling**

A great uncertainty is associated with the model initialization of soil carbon quality, as it was not measured, but obtained by matrix inversion with the assumption that the litter input has been the same for decades. Compared to total soil carbon stock, measuring soil carbon quality is much labour intensive and time-consuming. Moreover, data of soil carbon quality from different sources are partly or totally incompatible  due to the use of different chemical pools or protocols of fractionation (Blair et al., 1995). Therefore, measured data of soil carbon quality are generally lacking at worldwide scale. Such lack of information is a recurrent issue for soil carbon dynamics modeling (see Elliot et al. (1996), who has discussed the issue of "Measuring the modelable"). Many prevailing soil carbon models require setting carbon quality besides carbon quantity, e.g., Romul (Chertov et al., 2001), RothC (Coleman and Jenkinson, 1996), CENTURY versions Parton et al., 1987; Metherell et al., 1993, CBM-CFS3 (Kurz et al., 2009). Inappropriate setting of carbon quality in models may greatly change carbon stock predicts (Wutzler and Reichstein, 2007; Carvalhais et al., 2008; 2010).

In the present study, soil carbon quality data were unavailable at the French RENECOFOR sites. As a result, we used the simulated carbon quality at steady-state to feed Yasso07. This is a strong, but widely adopted assumption in soil carbon modelling work (Foereid et al., 2012). Alternative to the steady-state assumption, a relaxed equilibrium assumption has been  proposed (see Wutzler and Reichstein, 2007). The latter assumes that soil carbon pools (especially at sites that underwent disturbances in recent centuries) are not in steady-state, but in a transient state. At such a site, while the relatively labile pools (e.g., A, W, E and N pools in Yasso07) 
[revised manuscript text omitted]

[Figure]

Figure 6 Proportions of carbon pools (AWENH) at steady-state for all the RENECOFOR sites (*y*-axis) plotted against observed carbon stock at t1 until 0.4 m (*x*-axis). Each symbol represents one RENECOFOR site: green triangles stand for the sites dominated by conifers and orange dots stand for the sites dominated by broadleaves. For each boxplot, the lower and top edge of the box corresponds to the 25th and 75th percentile data points; lower and top bars the line within the box represents the median and the hollow points indicate outliers. Red letters below the boxplot denote the statistical diagnoses (t-test) with a significance level of *P*= 0.05*. No clear linear relationship was found between carbon quality and observed carbon stock at t1.

**Supplementary Materials**

**Supplementary Materials I:** Supplementary tables and figures.

**Supplementary Materials II:** Database for the meta-analysis of wood and litter chemical
composition.

---

## Author Response (AR2)

**General words to Referees.**

We truly thank both referees for their positive comments and highly valuable suggestions to improve the manuscript. We greatly appreciate their expertise. In this version, we erased the old traces from the two previous version. Only modification traces since the last version are left.

**Referee 1:**

**Referee**: Mao et al., submitted a revised version of their manuscript about the evaluation of the Yasso model over French forest. The paper has been well improved and the reading is now quite good.

**Authors**: Thank you for this positive remark to our manuscript!

**Referee**: Nevertheless I still disagree with the approach used to calculate the annual carbon stock changes for the simulation (eq. 7). I thought first it was a typo mistake in the first version of the manuscript but in their answers the authors confirmed that it was the good equation.

By doing ACCsim=(CSsim,t2-CSobs,t1)/(t2-t1) the author don't calculate the annual carbon stock changes in the simulation. Indeed, CSobs,t1 might be different than CSsim,t1 as shown by fig. S4 therefore if the error in the simulation of the carbon stocks at t1 is large the calculation of the ACC is mainly impact by this differences and not by the trends. Fig. S4 shown that Yasso tend to overestimate the carbon stocks for broadleaves whereas it underestimates the stocks for conifers. The impact for some points is probably not negligible. If you want to look to the capabilities of the model to reproduce the observed annual carbon stock changes ACCsim should be calculated as ACCsim=(CSsim,t2-CSsim,t1)/(t2-t1).

In my opinion this modification is mandatory before publication.

**Authors**: Thank you for re-raising this point. This indicates that we poorly explained this point in the previous version.

The concern raised by reviewer 1 probably comes from a misunderstanding of the initialization of the model. To estimate the change in soil carbon, the model was initialised based on the soil carbon stock observed at the first soil survey. We did not use the simulated carbon stock as it was obtained by supposing that the initial carbon stock was at steady state at one time (e.g. Ortiz et al., 2013). In our case, we have evidences that our soil systems were not at the steady state (Fig. S5).

This confusion is also probably due to the ambiguity of the newly added Fig. S5 (added after the R1 revision as demanded by the other reviewer, with the old name of "Fig. 5S"). Plotting "CS_sim at steady-state" against one of the two observed "CS_obs" (under the suggestion of Referee 2) should solely be regarded as a way to assess the disparity between theoretically obtained steady-state stock and the observed stock. We hope these explanations are now clearer.

As a result, we do not modify the equation and the associated results in the manuscript. Nevertheless, to improve the clarity regarding the reviewer's concerns we have (i) better justified the use of "CS_obs, t1" when presenting the Equation, see P15, LN10-18.; and (ii) clarified the captions of Figure S5 to make readers not think it is "CS_sim, t1", but "CS_sim at steady-state" , see Fig. S5.

**Referee 2:**

**Referee:** The revised paper acknowledges my main concern of improper initialization of carbon quality distribution by a short discussion. The authors argue that comparing another initialization method would be too much work and should go to a new paper.

However, I still think that the entire part on annual carbon accumulation (ACC) cannot be trusted with the currently applied assumption of initializing carbon quality distribution. This might be only my personal interpretation that I defend below, but the authors should better defend their assumption and the validity of their results. Or they should clearly acknowledge that only an upcoming paper will supply more reliably conclusions. For example, one of the conclusions is that its "Yasso07s failure of too much penalizing loss of labile carbon" (P21L4ff). I disagree. To my view it's the author's assumptions of scaling the more labile carbon stocks during initialization that determines the ACC pattern in the simulations and its missing correlation with observations.

**Authors**: Thank you for the understanding and also the paths you suggested.

Regarding the initialization method, on one side, we do not deny the existence of your concern, as the popular steady-state hypothesis (SSH) is indeed a strong one. On the other hand, we do not have intension to change the model's parameters or configuration as this is not the objective of this manuscript. So we continue to search compromises as follows.

First, we **added a new simulation which is based on an alternative initialization method** (but without modifying Yasso's configuration and parameter values). To initialize Yasso07, both the quantity and the quality of the soil carbon must be fixed. In the previous versions of the paper, the total quantity was fixed to the soil C stock measured at the first soil survey of the RENECOFOR. The quality was determined by estimating the C amounts at steady state for each chemical fraction (A, W, E, N, H) based on the spin-up/matrix conversion and by calculating their proportions in relation to their sum. These proportions were then applied to the observed C stock to split it in various pools. This approach does not consider the difference in carbon stability among these pools. This is however most likely that the fast-cycling pools such as A, W and E were at steady state at the first soil survey while the H pool could still be far from it (depending on the site history) (as suggested by the Referee). In the new version of the paper, we alternatively considered the C quantity obtained from the spin-up/matrix conversion for A, W, E and N and deduced the H amount by difference with the measured C stock. Although this initialization method is not perfect, it follows well the main idea of Wutzler (2007): due to possible disturbance, the fast-cycling AWEN pools and the slow-cycling H pool do not stabilize at the same time. The text representing this argument can be found in P12-13, Section 2.5.

Next, we **carried out a new sensitivity analysis** on the effect carbon quality (complete versus partial steady-states), crossed with initial carbon quantity (until 40 cm versus until 100 cm) and fine root:leaf ratios (from 0.1 to 4.0) on model fit. See the newly added Fig. S3 related to this analysis. We confirm that the alternative assumption (i.e., partial steady-states assumption, Fig. S3c and S3d) indeed gives better results than those given by complete steady-state assumption, i.e., the one we used in previous versions (Fig. S3a and S3b). In fact, when using CS until 0.4m to initialize Yasso07 (Fig. S3c), the model fit is the best. However, since we obtained the soil carbon stock data until 1.0 m from RENECOFOR from the last version of the manuscript, we decided to use CS until 1.0 for all the simulations related to the main text (Fig. S3d).

So in this new version, we have **redone all the stats and post-analyses** based on this new carbon quality assumption. Accordingly, since the results related to the complete steady-state assumption will not play a major role in the manuscript any more, we have deleted the sensitivity analysis 2.6.1 (effect of litter quality on the steady-state soil carbon quality), as well as its associated texts (e.g., old Section 3.3 and 4.4) and figures (including two supplementary figures), and also old Figure 6 (which was added in the last version). Instead, the Figures 3, 4 and 5 are updated. The new Fig.6 is served for illustrating the disparity of carbon quality between two assumptions. In Supple. Mat., Figure S5 is new (fit with complete steady-state assumption); Table 2, Figures S7 (boxplots on soil type), S8 (geographical distribution of residual signs) and S9 (PCA of residuals) are all updated. The section "sensitivity analyses" in Results has become 3.2, i.e., before the residual analyses (Sect. 3.3). Because we would like to start from general results and justify why we choose that particular case for residual analyses.

Then, as what you suggested, we've **continued to tune down the conclusion**: we also added a phrase in the perspective that expresses the meaning of "upcoming paper will supply more reliable conclusions" in P29, LN19-23. This prudence you suggested is totally ok and acceptable.

We decided to delete the speculative sentence "Yasso07s failure of too much penalizing loss of labile carbon…", as we didn't do a sensitivity analysis on the model's response as a function labile carbon quantity compared with other models (it is not the objective of the paper). See P22, LN16-21.

**Referee**: Thanks for providing Fig S4. From lines p14/22ff of the manuscript, I infer that „simulated" corresponds to the equilibrium carbon stocks and „observed" corresponds to measured 40cm stocks at initial time extrapolated down to 1m, correct? This should be clarified in the figure title. Moreover, mentioning ACC in the figure title is probably an error here.

**Authors**: We apologize for the unclearness of the text. In Fig. S5 (Fig. S4 in the old version), CS of both axes correspond to stock until 1 m. We modified the figure labels and the captions of Fig. S5 to make it clearer.

Yes, mentioning ACC is an error which is now corrected.

**Referee**: For the conifers it shows that observed stocks are larger than simulated equilibrium, hinting to some model inadequacy, or mismatches in input fluxes, or underestimate of stead state stocks, e.g. by a too high decomposition rate of the slow pool.

From comparing Fig S4 and Fig. 3 I see that where initial stocks are overestimated, also the stock change is overestimated, and similarly where initial stocks were underestimated, stock change is underestimated. To me, this hints to the suspected large effect of scaling faster pools when transferring steady state to observed stocks.

**Authors**: We don't deny for the case of conifers, possibly due to the inadaptability of model parameters to the dataset. Thank you for such a good reasoning.

The Fig. S5 (Fig. S4 in old version) also shows that for most broadleaved sites, observed stocks are lower than its steady-state equilibrium, indicating that equilibrium may not yet be reached at these sites.

Since the ACC fit with the alternative initialization method based on the "partial steady-state assumption" (by using simulated absolute values of AWE and the revised N and H, notably H) is improved (see the new Fig. 3) compared to that based on complete steady-state assumption (see the new Fig. S6 for a depth of 1.0 m). We suppose that, this method may, to some extent, mitigate the impact of such discrepancies: for broadleaves, the proportion of A+W+E (that gives rise more $CO_2$ away) are more enhanced than that at complete steady-state, reducing the model's overestimation of ACC at steady-state; for coniferous sites, the proportion of A+W+E will be pressed, reducing the model's underestimation of ACC at steady-state.

Such phenomena have been mentioned and discussed in Discussion, see P21, LN11-20, P21-22, LN30-2 and P23-24 Section 4.2.

Fig. 4 could probably also be explained by the same issue. When initial steady state stocks are higher than observed stocks, you downscale the faster pools, leading to stronger positive decadal ACC when pools develop towards steady state again. Similarly, when initial steady state stocks are lower, you upscale the faster pools leading to negative decadal ACC during simulation. The missing of this pattern in observations suggests to me to better not scale the faster pools.

**Authors**: See our responses above. With such an alternative assumption (while keeping model parameter/configuration untouched), such effect is mitigated. In this version, we've put the figure based on complete steady-state assumption model fit as Fig. S6 (down to 1.0 m this time). Now the Fig. 3 in the main text is based on the partial steady-state assumption (down to 1.0 m).

**Referee**: My conclusion from you results, therefore, is that probably the mismatch in initial distribution of qualities determines your pattern of carbon accumulation. To me most of the results come down to the assumption of keeping carbon quality distribution constant when adjusting stocks.

**Authors**: See our responses above too.

**Referee**: Specific comments:

I cannot get the message from Fig. S9. However, it should be crucial to my interpretation of importance of changes to steady state litter quality distribution for ACC. Can you simplify it?

**Authors**: We keep the Fig. S4 (Fig. S9 in old version) to give a general idea how and how much initial carbon quality can alter the model's output. In this version, the description on Fig. S4 in the main text has been simplified to 9 lines.

**Referee**: P22l15ff: Thanks for including discussion on the relaxed equilibrium assumption. Lack of information on modified rate of H pool should not be a problem at the time scale of interest. Just use a significantly slower decomposition rate, as suggested by Wutzler 2007. The difference will only be relevant at longer time scale.

**Authors**: See our response above.

**Referee**: P23L7: Note that the intended usage of depth-dependent decomposition rates renders the calibrated model only applicable at the site of calibration. Depth is often just a surrogate factor for other stabilization mechanisms. Trying to capture better indicators for these stabilization mechanisms is better than relying on their relation with depth that will change across sites.

**Authors**: Thank you for this comment and suggestion. We rephrased the sentence and added "capture better indicators for these stabilization mechanisms." See P25, LN7-8.

[revised manuscript text omitted]
 analysisWith a fixed initial soil carbon stock, we investigated the response of simulated final soil carbon quantity and quality to the setting 
[revised manuscript text omitted]

**Supporting Material I: Supplementary tables and figures**

**Table S1 Information on forest inventories for stand biomass estimation**

| Site | Dominant Species | Soil | Forest inventory for stand biomass | | | | Storm event (yr) | No. of thinnings |
|---|---|---|---|---|---|---|---|---|
| | | | Beginning(yr) | End (yr) | Span (yrs) | No. of inventories | | |
| QR_10 | *Quercus robur* | Calcisol | 1991 | 2009 | 18 | 7 | | 1 |
| QR_18 | *Quercus robur* | Planosol | 1991 | 2009 | 18 | 7 | | 3 |
| QR_40 | *Quercus robur* | Cambisol | 1992 | 2011 | 19 | 9 | 2009 | 2 |
| QR_49 | *Quercus robur* | Planosol | 1991 | 2010 | 19 | 6 | 2009 | 3 |
| QR_55 | *Quercus robur* | Calcisol | 1992 | 2009 | 17 | 6 | | 2 |
| QR_59 | *Quercus robur* | Luvisol | 1991 | 2010 | 19 | 8 | | 2 |
| QR_65 | *Quercus robur* | Cambisol | 1992 | 2012 | 20 | 6 | | 3 |
| QR_70 | *Quercus robur* | Luvisol | 1992 | 2011 | 19 | 7 | | 3 |
| QR_71 | *Quercus robur* | Luvisol | 1991 | 2009 | 18 | 6 | | 2 |
| QP_1 | *Quercus petraea* | Cambisol | 1991 | 2011 | 20 | 7 | | 2 |
| QP_3 | *Quercus petraea* | Cambisol | 1991 | 2009 | 18 | 7 | | 3 |
| QP_10 | *Quercus petraea* | Luvisol | 1991 | 2010 | 19 | 9 | | 3 |
| QP_18 | *Quercus petraea* | Luvisol | 1991 | 2009 | 18 | 7 | | 3 |
| QP_21 | *Quercus petraea* | Luvisol | 1991 | 2012 | 21 | 7 | | 2 |
| QP_27 | *Quercus petraea* | Luvisol | 1992 | 2009 | 17 | 9 | 1999 | 2 |
| QP_35 | *Quercus petraea* | Luvisol | 1991 | 2011 | 20 | 6 | | 2 |

| | | | | | | | | |
|---|---|---|---|---|---|---|---|---|
| QP_41 | *Quercus petraea* | Luvisol | 1991 | 2010 | 19 | 7 | | 1 |
| QP_51 | *Quercus petraea* | Cambisol | 1992 | 2004 | 12 | 5 | 1999 | 0 |
| QP_57a | *Quercus petraea* | Planosol | 1992 | 2009 | 17 | 8 | 1999 | 1 |
| QP_57b | *Quercus petraea* | Podzol | 1992 | 2009 | 17 | 5 | | 2 |
| QP_58 | *Quercus petraea* | Luvisol | 1991 | 2009 | 18 | 6 | | 3 |
| QP_60 | *Quercus petraea* | Planosol | 1992 | 2009 | 17 | 7 | | 2 |
| QP_61 | *Quercus petraea* | Luvisol | 1991 | 2009 | 18 | 7 | 1999 | 2 |
| QP_68 | *Quercus petraea* | Calcisol | 1992 | 2009 | 17 | 7 | 1999 | 2 |
| QP_72 | *Quercus petraea* | Luvisol | 1991 | 2009 | 18 | 8 | | 3 |
| QP_81 | *Quercus petraea* | Luvisol | 1992 | 2009 | 17 | 6 | | 1 |
| QP_86 | *Quercus petraea* | Luvisol | 1991 | 2009 | 18 | 6 | 1999 | 4 |
| QP_88 | *Quercus petraea* | Cambisol | 1992 | 2011 | 19 | 8 | | 3 |
| QP&QR_67 | *Quercus petraea & Q. robur* | Cambisol | 1992 | 2004 | 12 | 5 | 1999 | 3 |
| QP&QR_77 | *Quercus petraea & Q. robur* | Podzol | 1991 | 2009 | 18 | 6 | 1999 | 2 |
| PM_23 | *Pseudotsuga menziesii* | Cambisol | 1991 | 2008 | 17 | 7 | 1999 | 1 |
| PM_34 | *Pseudotsuga menziesii* | Cambisol | 1991 | 2010 | 19 | 7 | | 4 |
| PM_61 | *Pseudotsuga menziesii* | Luvisol | 1991 | 2011 | 20 | 7 | | 3 |
| PM_65 | *Pseudotsuga menziesii* | Cambisol | 1992 | 2004 | 12 | 5 | | 0 |

| | | | | | | | | |
|---|---|---|---|---|---|---|---|---|
| PM_69 | *Pseudotsuga menziesii* | Cambisol | 1991 | 2004 | 13 | 7 | 1999 | 1 |
| PM_71 | *Pseudotsuga menziesii* | Podzol | 1993 | 2013 | 20 | 11 | | 5 |
| PA_8 | *Picea abies* | Podzol | 1992 | 2009 | 17 | 5 | | 1 |
| PA_34 | *Picea abies* | Podzol | 1991 | 2009 | 18 | 7 | 2009 | 3 |
| PA_39a | *Picea abies* | Luvisol | 1991 | 2004 | 13 | 5 | | 1 |
| PA_63 | *Picea abies* | Andosol | 1991 | 2009 | 18 | 7 | | 3 |
| PA_71 | *Picea abies* | Podzol | 1991 | 2004 | 13 | 5 | | 1 |
| PA_73 | *Picea abies* | Cambisol | 1992 | 2007 | 15 | 5 | | 2 |
| PA_74 | *Picea abies* | Luvisol | 1991 | 2009 | 18 | 8 | | 3 |
| PA_81 | *Picea abies* | Podzol | 1992 | 2004 | 12 | 5 | | 1 |
| PA_87 | *Picea abies* | Podzol | 1991 | 2009 | 18 | 7 | 1999 | 2 |
| PA_88 | *Picea abies* | Cambisol | 1992 | 1999 | 7 | 3 | 1999 | 0 |
| FS_2 | *Fagus sylvatica* | Luvisol | 1992 | 2009 | 17 | 6 | | 2 |
| FS_3 | *Fagus sylvatica* | Cambisol | 1991 | 2009 | 18 | 8 | | 3 |
| FS_4 | *Fagus sylvatica* | Cambisol | 1992 | 2009 | 17 | 5 | | 0 |
| FS_9 | *Fagus sylvatica* | Podzol | 1992 | 2009 | 17 | 6 | | 1 |
| FS_14 | *Fagus sylvatica* | Cambisol | 1991 | 2013 | 22 | 8 | | 3 |
| FS_21 | *Fagus sylvatica* | Leptosol | 1991 | 2009 | 18 | 6 | 1999 | 1 |

| | | | | | | | | |
|---|---|---|---|---|---|---|---|---|
| FS_25 | *Fagus sylvatica* | Cambisol | 1991 | 2009 | 18 | 7 | | 3 |
| FS_26 | *Fagus sylvatica* | Leptosol | 1991 | 2009 | 18 | 6 | | 1 |
| FS_29 | *Fagus sylvatica* | Luvisol | 1991 | 2009 | 18 | 6 | | 3 |
| FS_30 | *Fagus sylvatica* | Podzol | 1991 | 2012 | 21 | 7 | | 2 |
| FS_52 | *Fagus sylvatica* | Leptosol | 1991 | 2005 | 14 | 6 | 1999 | 2 |
| FS_54a | *Fagus sylvatica* | Planosol | 1992 | 1999 | 7 | 3 | 1999 | 1 |
| FS_54b | *Fagus sylvatica* | Leptosol | 1992 | 1999 | 7 | 4 | 1999 | 0 |
| FS_55 | *Fagus sylvatica* | Podzol | 1992 | 2011 | 19 | 8 | 1999 | 2 |
| FS_60 | *Fagus sylvatica* | Luvisol | 1992 | 2009 | 17 | 7 | 1999 | 1 |
| FS_64 | *Fagus sylvatica* | Cambisol | 1992 | 2011 | 19 | 8 | | 3 |
| FS_65 | *Fagus sylvatica* | Cambisol | 1992 | 2009 | 17 | 7 | | 2 |
| FS_76 | *Fagus sylvatica* | Luvisol | 1991 | 2009 | 18 | 9 | | 2 |
| FS_81 | *Fagus sylvatica* | Podzol | 1992 | 2009 | 17 | 6 | | 1 |
| FS_88 | *Fagus sylvatica* | Cambisol | 1992 | 2009 | 17 | 8 | | 2 |
| LD_5 | *Larix deciduas* | Regosol | 1991 | 2014 | 23 | 6 | | 2 |
| PN_20 | *Pinus nigra* | Cambisol | 1991 | 2009 | 18 | 7 | | 2 |
| PN_41 | *Pinus nigra* | Podzol | 1991 | 2004 | 13 | 6 | 1999 | 2 |
| PP_17 | *Pinus pinaster* | Arenosol | 1991 | 2009 | 18 | 7 | 1999 | 1 |

| | | | | | | | | |
|---|---|---|---|---|---|---|---|---|
| PP_20 | *Pinus pinaster* | Cambisol | 1991 | 2004 | 13 | 5 | | 2 |
| PP_40a | *Pinus pinaster* | Podzol | 1992 | 2004 | 12 | 8 | | 2 |
| PP_40b | *Pinus pinaster* | Podzol | 1992 | 2009 | 17 | 7 | 2009 | 2 |
| PP_40c | *Pinus pinaster* | Podzol | 1992 | 2009 | 17 | 8 | 2009 | 3 |
| PP_72 | *Pinus pinaster* | Podzol | 1991 | 2010 | 19 | 7 | | 4 |
| PP_85 | *Pinus pinaster* | Arenosol | 1991 | 2011 | 20 | 6 | | 3 |
| PS_4 | *Pinus sylvestris* | Leptosol | 1991 | 2004 | 13 | 4 | | 0 |
| PS_15 | *Pinus sylvestris* | Cambisol | 1991 | 2011 | 20 | 7 | 1999 | 2 |
| PS_35 | *Pinus sylvestris* | Luvisol | 1991 | 2013 | 22 | 7 | | 3 |
| PS_41 | *Pinus sylvestris* | Podzol | 1991 | 2004 | 13 | 6 | 1999 | 2 |
| PS_44 | *Pinus sylvestris* | Luvisol | 1991 | 2010 | 19 | 7 | | 3 |
| PS_45 | *Pinus sylvestris* | Planosol | 1991 | 2005 | 14 | 7 | | 2 |
| PS_61 | *Pinus sylvestris* | Luvisol | 1991 | 1999 | 8 | 3 | 1999 | 0 |
| PS_63 | *Pinus sylvestris* | Cambisol | 1991 | 2009 | 18 | 7 | 1999 | 0 |
| PS_67a | *Pinus sylvestris* | Podzol | 1992 | 2009 | 17 | 7 | 1999 | 1 |
| PS_67b | *Pinus sylvestris* | Podzol | 1992 | 2013 | 21 | 9 | 1999 | 3 |
| PS_76 | *Pinus sylvestris* | Podzol | 1991 | 2009 | 18 | 7 | 1999 | 1 |
| PS_78 | *Pinus sylvestris* | Podzol | 1992 | 2007 | 15 | 6 | 1999 | 1 |

| ID | Species | Soil | Beginning (yr) | End (yr) | Mean span (yrs) | Mean no. of inventories | Frequency (storms/100 yrs) | Frequency (thinnings/10 yrs) |
|---|---|---|---|---|---|---|---|---|
| PS_88 | *Pinus sylvestris* | Podzol | 1992 | 2007 | 15 | 6 | 1999 | 1 |
| PS_89 | *Pinus sylvestris* | Podzol | 1991 | 1999 | 8 | 4 | 1999 | 1 |
| AA_5 | *Abies alba* | Cambisol | 1991 | 2009 | 18 | 5 | | 1 |
| AA_7 | *Abies alba* | Podzol | 1991 | 2010 | 19 | 6 | | 1 |
| AA_9 | *Abies alba* | Podzol | 1992 | 2008 | 16 | 7 | 2009 | 2 |
| AA_11 | *Abies alba* | Luvisol | 1992 | 2009 | 17 | 8 | | 2 |
| AA_25 | *Abies alba* | Cambisol | 1991 | 2012 | 21 | 8 | | 3 |
| AA_26 | *Abies alba* | Cambisol | 1991 | 2014 | 23 | 6 | | 2 |
| AA_38 | *Abies alba* | Cambisol | 1992 | 2009 | 17 | 6 | | 1 |
| AA_39 | *Abies alba* | Cambisol | 1991 | 2009 | 18 | 6 | | 2 |
| AA_57 | *Abies alba* | Cambisol | 1992 | 2009 | 17 | 9 | 1999 | 2 |
| AA_63 | *Abies alba* | Cambisol | 1992 | 2004 | 12 | 6 | | 1 |
| AA_68 | *Abies alba* | Cambisol | 1992 | 2012 | 20 | 7 | | 3 |
| All sites: | | | 1991 | 2014 | 17.0 | 6.6 | 2.1 | 1.1 |

**Table S2 Linear regressions for explaining the variability of annual carbon change residuals using soil physical and chemical properties**

| Variable | Broadleaves | | | | Conifers | | | |
|---|---|---|---|---|---|---|---|---|
| | $R^2$ | Slope | Intercept | *P*-value | $R^2$ | Slope | Intercept | *P*-value |
| Total nitrogen (in tN/ha) | 0.257 | -0.217 | 1.006 | <0.001 | 0.191 | -0.198 | 0.056 | <0.01** |
| Proportion of sand (in %) | 0.152 | 0.010 | -0.221 | <0.01** | 0.008 | 0.003 | -0.944 | >0.05 |
| Exchangeable Mg (in kmol/ha) | 0.138 | -0.011 | 0.255 | <0.01** | 0.000 | 0.000 | -0.761 | >0.05 |
| Exchangeable K (in kmol/ha) | 0.109 | -0.071 | 0.374 | <0.05* | 0.001 | 0.020 | -0.807 | >0.05 |
| Proportion of clay (in %) | 0.099 | -0.016 | 0.435 | <0.05* | 0.016 | -0.013 | -0.561 | >0.05 |
| Proportion of silt (in %) | 0.094 | -0.010 | 0.566 | <0.05* | 0.004 | -0.003 | -0.660 | >0.05 |
| Exchangeable Al (in kmol/ha) | 0.070 | -0.004 | 0.360 | >0.05 | 0.002 | -0.001 | -0.704 | >0.05 |
| Total phosphorus (in tN/ha) | 0.045 | -0.011 | 0.304 | >0.05 | 0.000 | 0.000 | -0.770 | >0.05 |
| Exchangeable Ca (in kmol/ha) | 0.016 | 0.000 | 0.135 | >0.05 | 0.004 | 0.000 | -0.729 | >0.05 |
| pH | 0.005 | 0.042 | -0.099 | >0.05 | 0.000 | 0.018 | -0.839 | >0.05 |
| Carbon:nitrogen ratio | 0.000 | 0.001 | 0.069 | >0.05 | 0.019 | 0.009 | -1.063 | >0.05 |

Note: the grey zone indicates the variables chosen for plotting the Figure 5 in the manuscript. $R^2$ = coefficient of determination;

[Figure]

Figure S1 Partitioning of soil carbon pools in Yasso07 (after Tuomi et al., (2011b)) Letters: A: hydrolysable in Acid; W: soluble in Water; E: soluble Ethanol; N: Non-soluble; H: recalcitrant Humus. Solid arrows represented the carbon flows that are statistically significant from zero. Dashed arrows refer to the carbon flows toward H. Values in each pool is an example inverse of mean residence time (*1/k*, in year) estimated using Yasso07 parameters.

[Figure]

Figure S23 Distributions of fine root:foliage ratio of litter input in different tree functional types calculated using the equation of Raich and Nadelhoffer (1989), see Jonard et al., (2017). For each boxplot, the lower and top edge of the box corresponds to the 25[th] and 75[th] percentile data points; lower and top bars the line within the box represents the median and the hollow points indicate outliers. Median values are shown beside median lines. "X" indicates mean values: $1.08 \pm 0.02$ (mean $\pm$ standard error) for sites dominated by conifers and $1.01 \pm 0.02$ for sites dominated by broadleaves.

[Figure]

Figure S32 Influence of the choice of model initialization method for soil carbon quantity (stock until 0.4 m versus stock until 1.0 m) and quality (complete versus partial steady-state assumption) and the choice of  fine root:foliage ratio of litter input (from 0.1 to 4.0) on the performance of Yasso07 toward the French RENECOFOR data. RMSE – root mean square error; Error bars are standard deviations of 10 simulations differing in parameters which were randomly chosen. Red dash lines perpendicular to *x*-axis: the two thin ones showing the values of fine root:foliage ratio for the minima of RMSE for broadleaves (0.1) and conifers (1.9), respectively; the thick red dash line at 1.0 (i.e., the ratio used for result presentation) showing that the RMSE of broadleaves and conifers are slightly higher than the minima, but still acceptable. The case in (c) gives the best model fit (lowest RMSE), but the case in (d) was preferentially chosen, as Yasso07 is validated by and predicts soil carbon data until 1.0 m.

[Figure]

Figure S48 Sensitivity analysis of the impact of carbon pool composition of initial soil C stock (*x*-axis (↙), in %) and simulation length (*y*-axis (→), in logarithmic years) on final soil carbon stock (*z*-axis (↑), in tC ha$^{-1}$). Here, the results are generated using the mean broadleaved litter input quantity and quality of the RENECOFOR sites. Initial soil carbon stock was fixed to 100 tC ha$^{-1}$. Subplots in each row show the final stock evolution of one type of soil carbon pools (i.e. A, W, E, N and H). Particularly, in the 2$^{nd}$ row W and E were combined due to their weak quantities in most of cases. Subplots in each column show the effect of one type of soil chemical groups on the final stocks of the five soil carbon pools (each of them for the first four and the last one is the total stock). In each subplot, a membrane (with grids for three-dimensional effect) represents the loess fit (polynomial equation) to *z* (in tC ha$^{-1}$) as a function of *x* and *y*; the color of the membrane represent the relative value of *z* (in %), i.e. the proportion of one soil carbon pool within the total soil carbon stock. No color is assigned to the membranes in the last row, because the relative value is 100 %. Blue lollipops denote the standard deviations of the simulated mean z (on the membrane surface) given each (x, y) locations, which follow a systematic distribution.

[Figure]

Figure S4 S5 Comparing the ison between simulated and observed annual steady-state carbon stock until 1 m (CS, in tC ha$^{-1}$) with the observed carbon stock until 1 m at t1, which were used for model input. Round and triangle symbols represent sites dominated by broadleaves and conifers, respectively. The chosen fine root:foliage ratio for broadleaves and conifers is 1.0. Error bars represent standard errors; hollow and filled points represent non-significant and significant differences between simulated and observed ACC according to t-test (at 95% confidence level).

[Figure]

Figure S6 Comparison between simulated and observed annual carbon stock changes (ACC, in tC ha$^{-1}$ year$^{-1}$). Round and triangle symbols represent sites dominated by broadleaves and conifers, respectively. The complete steady-state assumption was used for initializing carbon quality of the stock until 1.0 m. The chosen fine root:foliage ratio for broadleaves and conifers is 1.0. To facilitate discussions, we set Roman numbers (I-VI) denoting the six zones in which data points are distributed. In (a), error bars represent standard errors; hollow and filled points represent non-significant and significant differences between simulated and observed ACC according to t-test (at 95% confidence level). In (b), case of significance: 1 – no significant difference from 0 for neither observed nor simulated ACC; 2 - a significant difference from 0 for either observed or simulated ACC and 3: - a significant difference from 0 for both observed and simulated ACC.

[Figure]

Figure S75 Distributions of the residuals (simulated minus observed annual carbon stock changes) of Yasso07's fit for sites dominated by conifers. For each boxplot, the lower and top edge of the box corresponds to the 25th and 75th percentile data points; lower and top bars the line within the box represents the median; no outilier points in this case. "X" indicates mean values: -0.33 ± 0.15 (mean ± standard error) for cambisol,-0.41 ± 0.32 for luvisol and -1.20 ± 0.28 for podzol. Species accronyms: AA – *Abies alba*; PM – *Pseudotzuga menziesii*; PA – *Picea abies*; PS – *Pinus sylvestris*; PN – *Pinus nigra*; PP – *Pinus pinaster*. Letters above boxplots indicate diagnostics according to Tukey HSD test. Colors for different species: deep red for species that can be found for all the three types of soil; blue for species that can only found for cambisol and podzol, but not for luvisol.

[Figure]

Figure S6 S8 Spatial visualization of residuals (i.e. the difference between simulated and observed annual carbon changes) for sites dominated by broadleaves (a) and conifers (b). Colors: red – overestimation with residuals being significantly > 0; blue – underestimation with residual being significantly < 0; grey – residuals that are not significantly different from 0. Species abbreviations: FS – *Fagus sylvatica*; QP– *Quercus petraea*; QR - *Quercus robur* (*including* two mixed *Quercus* sites); PS - *Pinus sylvestris*; AA- *Abies alba*; PA - *Picea abies*.

[Figure]

Figure S97 Relationships among indicators of site features and model predicts using principal component analyses for sites dominated by conifers (a) and broadleaves (b), respectively. Colours of arrows: red – residuals of annual carbon change and observed initial carbon stock; grey – soil physical and chemical properties; blue –site geographical and climatic variables. Each point corresponds to one RENECOFOR site. See Table S2 for full names of soil properties.

---

## Author Response (AR3)

**General words.**

We truly thank the Section Editor for the Accept decision based on our last version.

As suggested, in this current version: a thorough linguistic check has been done by a native English speaker. We've also corrected the three minor language suggestions given by Referee 1.

There is no modification on any results or conclusions since last version.

We have erased the old traces from the previous versions. Only modification traces (linguistic check) since the last version are left.

**Referee 1:**

**Thank you for adopting and comparing the partial steady state initialization. I can follow the conclusions now. The paper reads still quite lengthy but I do not have specific suggestions how to improve. I just report a few minor language suggestions:**

Authors: Thank you !

**p19 l 3 typo: "c.a." You do not need the acronym. I suggest using "approximately" or "about"**

Authors: Done

**p22 l 24: Suggest replacing "standing for" with "representing"**

Authors: Done

**p23 K 15 a "more" is missing after "much"**

Authors: Done

**Referee 2 :**

**The point that was still problematic was better explained. I now understand better and the calculation of ACCsim is correct.**

Authors: Thank you !